 eLIFE

# Decoding the neural mechanisms of human tool use

**Jason P Gallivan[1,2]\*, D Adam McLean[3], Kenneth F Valyear[4], Jody C Culham[3,5,6]**

[1]Department of Psychology, Queen's University, Kingston, Canada; [2]Centre for Neuroscience Studies, Queen's University, Kingston, Canada; [3]Brain and Mind Institute, Natural Sciences Centre, University of Western Ontario, London, Canada; [4]Department of Psychological Sciences, Brain Imaging Center, University of Missouri, Columbia, United States; [5]Department of Psychology, University of Western Ontario, London, Canada; [6]Neuroscience Program, University of Western Ontario, London, Canada

**Abstract** Sophisticated tool use is a defining characteristic of the primate species but how is it supported by the brain, particularly the human brain? Here we show, using functional MRI and pattern classification methods, that tool use is subserved by multiple distributed action-centred neural representations that are both shared with and distinct from those of the hand. In areas of frontoparietal cortex we found a common representation for planned hand- and tool-related actions. In contrast, in parietal and occipitotemporal regions implicated in hand actions and body perception we found that coding remained selectively linked to upcoming actions of the hand whereas in parietal and occipitotemporal regions implicated in tool-related processing the coding remained selectively linked to upcoming actions of the tool. The highly specialized and hierarchical nature of this coding suggests that hand- and tool-related actions are represented separately at earlier levels of sensorimotor processing before becoming integrated in frontoparietal cortex.

**\*For correspondence:** jasongallivan@gmail.com

**Reviewing editor**: Dora Angelaki, Baylor College of Medicine, United States

## Introduction

Tool use, whether using a stone, stick, rake, or pliers, provides an extension of the body (*Van Lawick-Goodall, 1970*) and involves, among other things, the transfer of a proximal movement goal for the hand into a more distal goal for the tool (*Johnson and Grafton, 2003*; *Arbib et al., 2009*). A compelling demonstration that this transfer might actually occur at the cortical level comes from neural recordings of grasping neurons in the ventral premotor cortex (PMv) and motor cortex (M1) of macaque monkeys trained to use pliers (*Umilta et al., 2008*). In both these areas, many neurons that encoded the specifics of hand grasping subsequently encoded tool grasping, even when use of the specific tool (reverse pliers that close as the hand grip opens) required hand kinematics opposite to those required when grasping with the hand alone. These findings suggest that tool use is supported by an effector-independent level of representation, in which the overall goal of the motor act is coded separately from the precise hand kinematics required to operate the tool. In further support of this notion, findings from human neuropsychology (*Berti and Frassinetti, 2000*; *Maravita and Iriki, 2004*), human behavior (*Gentilucci et al., 2004*; *Cardinali et al., 2009*, *2012*), and macaque monkey neurophysiology (*Iriki et al., 1996*) suggest that following training, a tool may actually become incorporated into the body schema of the actor and coded as an extension of the hand/limb. While provocative, how well does this single mechanism explain the neural substrates of tool use in humans, particularly within established networks that have been identified for tools (*Lewis, 2006*), hand actions (*Culham et al., 2006*), and body perception (*Peelen and Downing, 2007*)?

**eLife digest** The use of tools is a key characteristic of primates. Chimpanzees—our closest living relatives—use sticks to probe for termites as well as stones to crack open nuts, and have even been seen using specially sharpened sticks as spear-like tools for hunting. However, despite its importance in human evolution, relatively little is known about how tool use is supported by the brain.

One possibility is that the brain areas involved in controlling hand movements may also begin to incorporate the use of tools. Another is that distinct brain areas evolved to enable tool use. To test these ideas, Gallivan et al. scanned the brains of human subjects as they reached towards and grasped an object using either their right hand or a set of tongs. The tongs had been designed so that they opened whenever the subjects closed their grip, thereby requiring subjects to perform a different set of movements to use the tongs as opposed to their hand alone.

Three distinct patterns of brain activity were observed. First, areas previously linked to the processing of hand movements and the human body were found to represent actions of the hand alone (and not those of the tool), whereas areas previously linked to the processing of tools and tool-related actions represented actions of the tool alone (and not those of the hand). Second, areas of motor cortex implicated in the generation of movement represented actions performed with both the hand and the tool, but showed distinct activity patterns according to which of these was to be used.

Lastly, areas associated with high-level cognitive and action-related processing showed similar patterns of activity regardless of whether the subjects were about to use the tongs or just their hand. Given that use of the hand and tool required distinct patterns of muscle contractions, this suggests that these higher-level brain regions must be encoding the action itself rather than the movements needed to achieve it.

This study is one of the first to use functional neuroimaging to examine real as opposed to simulated tool use, and increases our understanding of the neural basis of tool use in humans. This knowledge could ultimately have applications for the development of brain-machine interfaces, in which electrodes implanted in motor regions of the brain are used to control prosthetic limbs.

Although considerable research has been done on the brain networks specialized for visual processing of tools and bodies and the visual-motor processing of hand actions, these topics have largely been studied in isolation. Increasing evidence from functional magnetic resonance imaging (fMRI) suggests that human frontoparietal and occipitotemporal cortex contain specialized regions that selectively represent tools and bodies (*Downing et al., 2001*; *Lewis, 2006*; *Frey, 2007*; *Peelen and Downing, 2007*; *Peeters et al., 2009*; *Valyear and Culham, 2010*; *Bracci et al., 2012*). For instance, when individuals view, imagine, or pantomime tool use actions, the supramarginal gyrus (SMG), posterior middle temporal gyrus (pMTG), and dorsal premotor cortex (PMd)—areas that have shown some of the greatest evolutionary expansion in humans (*Van Essen and Dierker, 2007*)—are often co-activated (*Lewis, 2006*; *Frey, 2007*). fMRI studies further suggest that human occipitotemporal cortex also contains body-selective regions for perception, such as the extrastriate body area (EBA), which preferentially respond to viewing of the body and its parts (*Astafiev et al., 2004*; *David et al., 2007*; *Peelen and Downing, 2007*). The frontoparietal regions activated by tools are spatially close to (and perhaps overlapping with) brain areas implicated in hand actions, particularly the grip component of reach-to-grasp actions. Specifically, SMG lies very near the grasp-selective anterior intraparietal sulcus (aIPS, *Chao and Martin, 2000*; *Valyear et al., 2007*) and PMd shows grasp-selective as well as tool-selective responses (*Grezes and Decety, 2002*; *Gallivan et al., 2011*). In addition, real hand actions activate other frontoparietal regions including the superior parieto-occipital cortex (SPOC) region, PMd, and additional areas along the IPS (*Culham et al., 2006*; *Filimon, 2010*), but the specific role of these areas in tool use remains unexplored. Moreover, almost all of the human neuroimaging studies of tools to date have used proxies for real tool use (reviewed in *Lewis, 2006*), including visual stimuli such as images or movies (e.g., *Beauchamp et al., 2002*), semantic tasks (e.g., *Martin et al., 1995*), or simulated tool actions like pantomiming, imitating or imagining tool use (e.g., *Johnson-Frey et al., 2005*; *Rumiati et al., 2004*) or making perceptual judgments about how one would use a tool

(e.g., *Jacobs et al., 2010*). It remains unclear whether the highly specialized brain areas within these tool-, body-, and action-related networks in humans also play important roles in planning real movements with a tool or with the body (hand) alone.

The purpose of the current study was to examine exactly how and where in the human brain tool-specific, hand-specific, and effector-independent (shared hand and tool) representations are coded. To this aim we used fMRI to examine neural activity while human subjects performed a delayed-movement task that required grasp or reach actions towards a single target object. Critically, subjects performed these two different movements using either their hand or reverse tongs, which required opposite operating kinematics compared to when the hand was used alone. This manipulation allowed us to maintain a common set of actions throughout the experiment (grasping vs reaching) while at the same time varying the movement kinematics required to achieve those actions (i.e., depending on whether the hand vs tool effector was used). Using multi-voxel pattern analysis (MVPA) to decode preparatory (pre-movement) signals, we then probed exactly where in frontoparietal cortex and in tool- and body-selective areas in occipitotemporal cortex movement plans (grasping vs reaching) for the hand and tool were distinct (effector-specific) vs where signals related to upcoming actions of the hand could be used to predict the same actions performed with the tool (effector-independent).

Consistent with an effector-specific coding of hand- and tool-related movements we found that preparatory signals in SPOC and EBA differentiated upcoming movements of the hand only (i.e., hand-specific) whereas in SMG and pMTG they discriminated upcoming movements of the tool only (i.e., tool-specific). In addition, in anterior parietal regions (e.g., aIPS) and motor cortex we found that pre-movement activity patterns discriminated planned actions of 'both' the hand and tool but, importantly, could not be used to predict upcoming actions of the other effector. Instead, we found that this effector-independent type of coding was constrained to the preparatory signals of a subset of frontoparietal areas (posterior IPS and premotor cortex), suggesting that in these regions neural representations are more tightly linked to the goal of the action (grasping vs reaching) rather than the specific hand movements required to implement those goals.

## Results

fMRI (3 Tesla) was used to measure the blood oxygenation level-dependent (BOLD) signal in the brains of 13 right-handed subjects (7 females; mean age: 25.7 years) during a slow event-related design with a delay interval. Subjects used either the right hand or a tool (controlled by the right hand) to execute a precision reach-to-grasp (Grasp) or reach-to-touch (Reach) movement towards a single centrally located real three-dimensional (3D) target object made of Lego blocks (*Figure 1B*). The tool used was a set of reverse tongs; when the hand closed on the grips, the ends of the tongs would open and vice versa. As such, different hand kinematics were required to operate the tool compared to when the hand was used alone. Use of the hand and tool were alternated across experimental runs. The position of the target object was changed between hand and tool experimental runs in order for the grasps and reaches to be performed at a comfortable distance for each effector (*Figure 1B*). On each trial, subjects were first cued to the action to be carried out (grasp or reach). Then, following a delay period, they performed the instructed action (with the hand or tool, depending on the experimental run). The delay timing of the paradigm allowed us to divide the trial into discrete time epochs and isolate the sustained plan-related neural responses that evolve prior to movement from the transient visual response (Preview phase) and the movement execution response (Execute phase; *Figure 1C,D*).

We implemented MVPA in specific frontoparietal and occipitotemporal cortex regions-of-interest (ROIs) for each time-point within a trial and examined, during movement planning (Plan Phase): 1) whether we could predict upcoming grasps (G) vs reaches (R) with either the hand (i.e., Hand-G vs Hand-R) or tool (i.e., Tool-G vs Tool-R) or both and 2) where in the network of areas preparatory patterns of activity for the hand could be used to predict preparatory patterns of activity for the tool and vice versa (e.g., where Hand-G predicts Tool-G activity, and vice versa). With respect to this second aim, it is important to note that based on differences between hand and tool experimental runs, a brain area showing effector-independent preparatory activity patterns cannot be attributable to low-level similarities in motor kinematics (i.e., because the hand and tool required opposite operating mechanics) or sensory input across trial types (i.e., because the object's visual position with respect to fixation changed between hand and tool runs).

We first localized a common set of action-related ROIs within each individual subject for subsequent MVPA. These ROIs were defined by performing a whole-brain voxel-wise search contrasting the

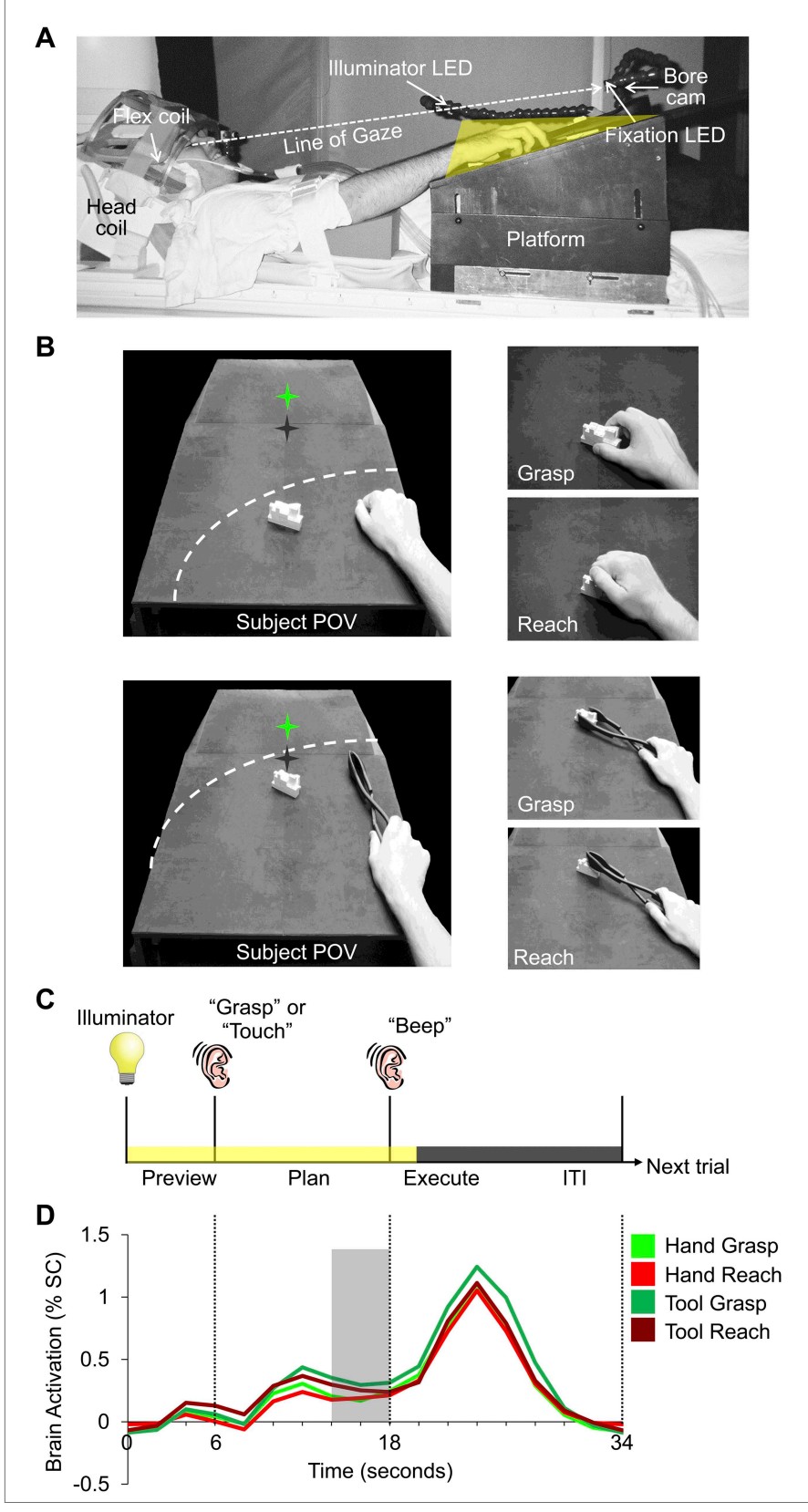

**Figure 1**. Experimental methods and evoked neural activity. (**A**) Subject setup shown from side view. (**B**) (Left) experimental apparatus and target object shown from the subject's point of view for experimental runs where
*Figure 1. Continued on next page*

*Figure 1. Continued*

either the hand (top) or reverse tool (bottom) were used. The location of the target object (white block) was switched between run types but did not change its position from trial-to-trial within a imaging run. Dashed line represents the participant's arc of reachability for each run type. In both cases (left panels), the hand is shown at its starting location. Green star with dark shadow represents the fixation LED and its location in depth. (Right) Hand and tool positions during movements performed by the subject. (**C**) Timing of each event-related trial. Trials began with the 3D object being illuminated while the subject maintained fixation (Preview phase; 6 s). Subjects were then instructed via headphones to perform one of two movements: Grasp the object ('Grasp') without lifting it or Touch the object ('Touch'), initiating the Plan phase portion of the trial. Following a fixed delay interval (12 s), subjects were cued (by an auditory 'beep') to perform the instructed movement (initiating the Execute phase) and then return to the starting location. 2 s after the Go cue, vision of the workspace was extinguished and participants waited for the following trial to begin (14-s intertrial interval, ITI). (**D**) Averaged fMRI activity from left dorsal premotor (PMd) cortex, time-locked to trial length. MVPA was performed using single fMRI trials in two ways: 1) based on the % signal change (SC) BOLD activation evoked for each single time point in the trial (time-resolved decoding), allowing us to pinpoint when predictive movement information was available and 2) based on a windowed average of the % SC BOLD activation in the 4 s (2 imaging volumes) prior to movement initiation (denoted by the gray shaded bar).

activity evoked during movement generation (i.e., movement planning [Plan phase] and execution [Execute phase]) vs the activity evoked during simple visual object presentation (Preview phase; when subjects had vision of the target object yet were unaware of which action [Grasp vs Reach] to perform). This [Plan & Execute > 2*Preview] contrast revealed activity throughout a well-documented frontoparietal network of areas (*Figure 2* and *Table 1* for coordinates). Within this network, we focused MVPA on 10 commonly described neuroanatomical ROIs in the left hemisphere (contralateral to the arm used), each previously implicated in either hand-related and/or tool-related processing. We localized superior parieto-occipital cortex (SPOC), posterior intraparietal sulcus (pIPS), middle IPS (midIPS), motor cortex, and dorsal premotor (PMd) cortex—a group of well-known parietal and frontal areas generally implicated in hand-related movement planning processes in human and/or macaque cortex (*Culham et al., 2006*; *Andersen and Cui, 2009*; *Cisek and Kalaska, 2010*). In addition, we defined the anterior intraparietal sulcus (aIPS), a region just posterior to aIPS (post. aIPS), supramarginal gyrus (SMG), and ventral premotor (PMv) cortex—a group of parietal and frontal areas generally implicated in hand preshaping and tool-related processes (*Rizzolatti and Luppino, 2001*; *Culham et al., 2006*; *Lewis, 2006*; *Umilta et al., 2008*). One additional area, left somatosensory (SS-cortex), was selected as a sensory control region, not expected to accurately decode movements until stimulation of the hand's mechanoreceptors at movement onset (i.e., at the Execute phase).

A subset of participants from the motor experiment (n = 8) were recruited for a second localizer experiment. The purpose of this second scan session was to localize well-documented visual-perceptual ROIs involved in body- and tool-selective processing and then examine, using the motor experiment data from these same subjects, whether hand and tool movement plans could be decoded from these separately identified object-selective areas. One area of interest, EBA, has been widely implicated in coding perceptual functions related to the body, including the identification of other individuals, the processing of others' emotional states and movements, as well as perception of the self and representations of the body schema (*Downing and Peelen, 2011*). Likewise, pMTG has been implicated in coding numerous perceptual functions related to tools, including tool motion, identification, naming, auditory processing, and the retrieval of semantic information concerning their use (e.g., *Martin et al., 1996*; *Beauchamp and Martin, 2007*; *Lewis et al., 2005*). Interestingly, both ROIs are also activated by self-generated movements (unseen hand actions in the case of EBA and pantomimed tool actions in the case of pMTG; *Lewis, 2006*; *Frey, 2007*; *Downing and Peelen, 2011*). The aim here was to clarify the nature and specificity of these sensory-motor responses in the context of planning object-directed hand and tool movements.

To localize these visual-perceptual ROIs we used a standard block-design localizer task in which participants were required to view static color photos of familiar tools, headless bodies, non-tool objects, and scrambled up versions of these same stimuli (*Figure 5A* for timing and protocol; see also *Valyear and Culham, 2010*). To identify the brain areas selectively involved in tool-related visual processing, in each subject we searched for regions showing heightened activation for tools compared to

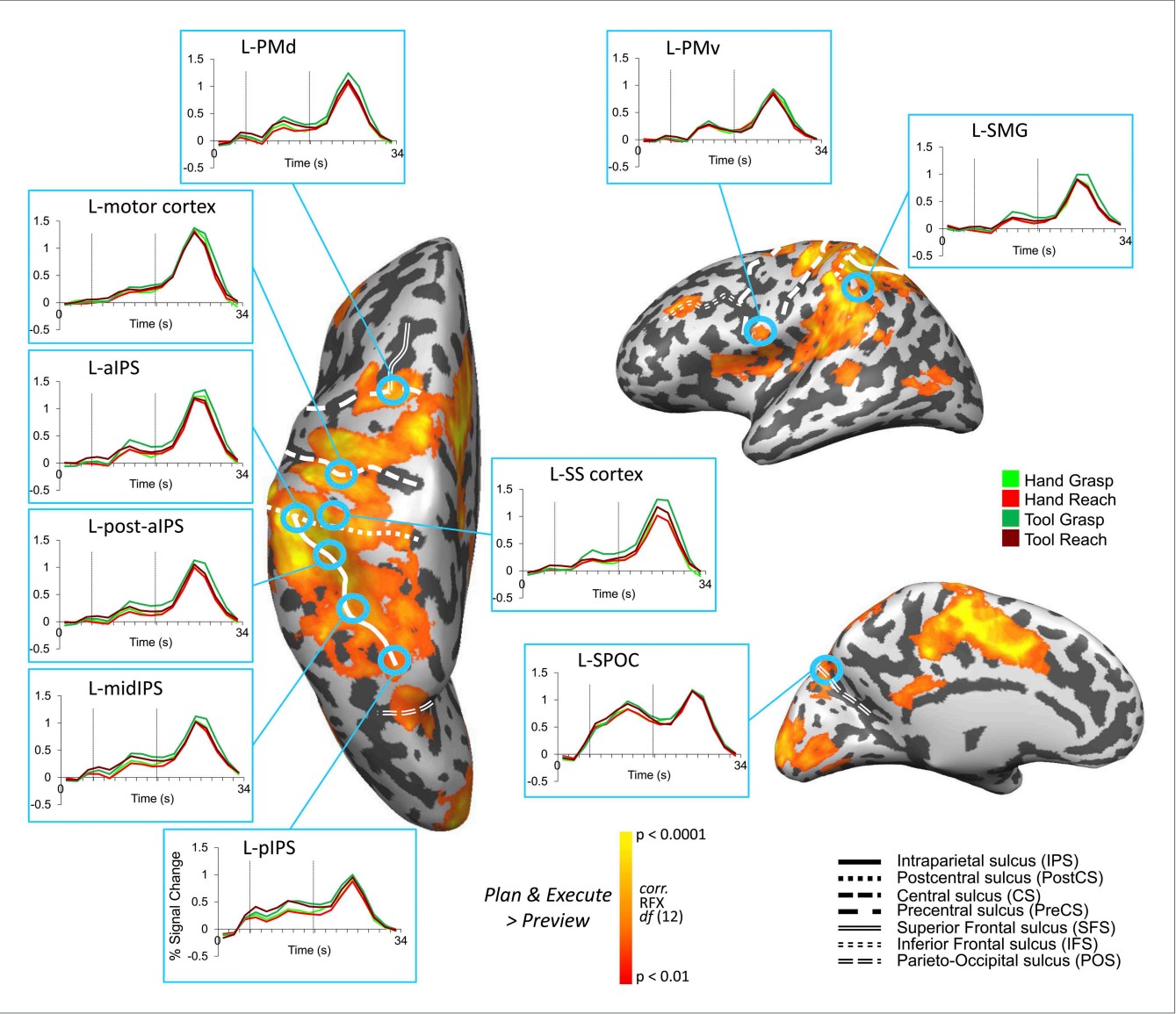

**Figure 2**. Frontoparietal brain areas selected for movement plan decoding. Cortical areas that exhibited larger responses during movement preparation and/or execution than the preceding visual phase [(Plan + Execute) > 2*(Preview)] are shown as orange/yellow activation. Results calculated across all subjects (Random Effects GLM) are displayed on one representative subject's inflated cortical hemispheres. The general locations of the selected ROIs are outlined in circles (actual ROIs were anatomically defined separately in each subject). Linked to each ROI is the corresponding % SC BOLD activity averaged across voxels, trials, and subjects within each ROI and plotted according to trial length. This time course activity clearly delineates the sustained preparatory responses that form prior to movement onset in each area. Vertical lines correspond to the onset of the Plan and Execute phases of each trial (from left to right). Sulcal landmarks are denoted by white lines (stylized according to the corresponding legend). ROI acronyms are spelled out in main text.

headless bodies, non-tool objects, and scrambled stimuli (conjunction contrast: [(Tools > Bodies) AND (Tools > Objects) AND (Tools > Scrambled)], t = 3, p<0.005, corrected). Across subjects, this contrast revealed consistent activity in two brain regions: 1) pMTG and 2) an area located anteriorly along the IPS (tool-aIPS, t-aIPS). The anatomical locations of these tool-specific activations were highly consistent with previous investigations (**Figures 5 and 6**; **Mahon et al., 2007**; **Valyear and Culham, 2010**). We next searched for body-selective regions by contrasting activity for bodies compared to tools, non-tool objects, and scrambled stimuli (conjunction contrast: [(Bodies > Tools) AND (Bodies > Objects) AND (Bodies > Scrambled)], t = 3, p<0.005, corrected). Across subjects, this contrast revealed consistent activity in EBA, which responds selectively to human bodies and body parts when compared

**Table 1.** ROIs with corresponding Talairach coordinates (mean x, y, and z centre of mass and standard deviations) and sizes

| ROI name | Tailarach coordinates | | | | | | ROI size | |
|---|---|---|---|---|---|---|---|---|
| | x | y | z | std x | std y | std z | mm³ | Nr voxels |
| Parieto-frontal ROIs | | | | | | | | |
| L Superior parieto-occipital cortex (SPOC) | −8 | −75 | 29 | 2.4 | 2.9 | 3.6 | 1469 | 54 |
| L Posterior intraparietal sulcus (pIPS) | −22 | −68 | 45 | 3.4 | 3.7 | 4.1 | 1640 | 61 |
| L Middle intraparietal sulcus (midIPS) | −32 | −56 | 46 | 4.4 | 4 | 3.9 | 1943 | 72 |
| L Posterior anterior intraparietal sulcus (post. aIPS) | −42 | −49 | 43 | 3.6 | 4.3 | 3.9 | 2290 | 85 |
| L Anterior intraparietal sulcus (aIPS) | −42 | −40 | 42 | 4.2 | 4.3 | 4.2 | 2067 | 77 |
| L Supramarginal gyrus (SMG) | −56 | −35 | 33 | 3.7 | 3.7 | 4 | 1479 | 55 |
| L Motor cortex | −38 | −29 | 48 | 4.3 | 4.3 | 4.3 | 2407 | 89 |
| L Dorsal premotor (PMd) cortex | −26 | −14 | 52 | 4.2 | 3.8 | 4.1 | 2135 | 79 |
| L Ventral premotor (PMv) cortex | −52 | 3 | 15 | 3.3 | 3.8 | 3.1 | 1460 | 54 |
| L Somatosensory (SS) cortex | −39 | −40 | 48 | 2.8 | 2.8 | 2.9 | 1592 | 59 |
| Localizer-defined ROIs | | | | | | | | |
| L tool-anterior intraparietal sulcus (t-aIPS) | −41 | −42 | 46 | 3.3 | 3.4 | 4.2 | 1038 | 38 |
| L Posterior middle temporal gyrus (pMTG) | −53 | −57 | −3 | 4.1 | 3.1 | 3.3 | 621 | 23 |
| L Extrastriate body area (EBA) | −49 | −72 | 1 | 2.8 | 3.6 | 3.7 | 851 | 32 |

Mean ROI sizes across subjects from ACPC data (in mm³ and functional voxels). Std = standard deviation.

with objects and other control stimuli (*Downing et al., 2001*; *Figure 6*). Notably, in each subject we found a highly consistent spatial relationship between EBA and pMTG: pMTG was characteristically positioned lateral, ventral, and anterior to EBA (consistent with *Valyear and Culham, 2010*).

## Movement plan decoding

To pinpoint when predictive movement information was available in the spatial voxel patterns, we ran a single-trial decoding analysis for each point in time over the course of a trial (*Soon et al., 2008*; *Harrison and Tong, 2009*). This time-resolved decoding analysis revealed a full range of decoding profiles during movement planning across the network of specified regions. For instance, preparatory voxel patterns in SPOC and EBA accurately predicted upcoming grasping vs reaching actions with the hand only whereas preparatory voxel patterns in SMG and pMTG successfully predicted grasping vs reaching actions with the tool only (*Figures 3 and 6*, red and blue decoding traces). Notably, in nearly all the remaining regions, we were able to successfully use the activity patterns to predict the action performed (grasping vs reaching) for both the hand and tool effector (*Figures 3–5*, red and blue decoding traces; purple traces will be discussed in the next section entitled 'Separate and shared representations for the hand and tool'). For instance, in parietal cortex, preparatory activity in pIPS, midIPS, post. aIPS, aIPS, and t-aIPS could be used to accurately discriminate which object-directed hand or tool movement was to be performed moments later (for overlap between t-aIPS and the post. aIPS and aIPS regions, see *Figure 5*). Likewise, in frontal cortex, predictive movement activity for hand and tool actions was also found in motor cortex, PMd, and PMv. Importantly, consistent with expectations and previous investigations (*Gallivan et al., 2011a, 2011b*), our sensory control region, SS-cortex, failed to decode any planned movements and only discriminated the different actions upon execution (*Figure 3*). To verify the observations obtained from the time-resolved decoding analysis, we also averaged the spatial activity patterns generated over a 4-s (2-volume) window of time immediately prior to the cue for subjects to perform the movement (denoted by gray shaded bars in *Figures 3–6*). In line with our previous fMRI investigations (*Gallivan et al., 2011a, 2011b*), this plan-epoch decoding approach supports the notion that discriminatory predictive signals for movement can arise moments prior to action execution.

To ensure our decoding accuracies could not result from spurious factors (e.g., task-correlated head or arm movements), we ran the same classification analyses in two non-brain ROIs where decoding should not be expected: the right ventricle and outside the brain. Consistent with our previous work

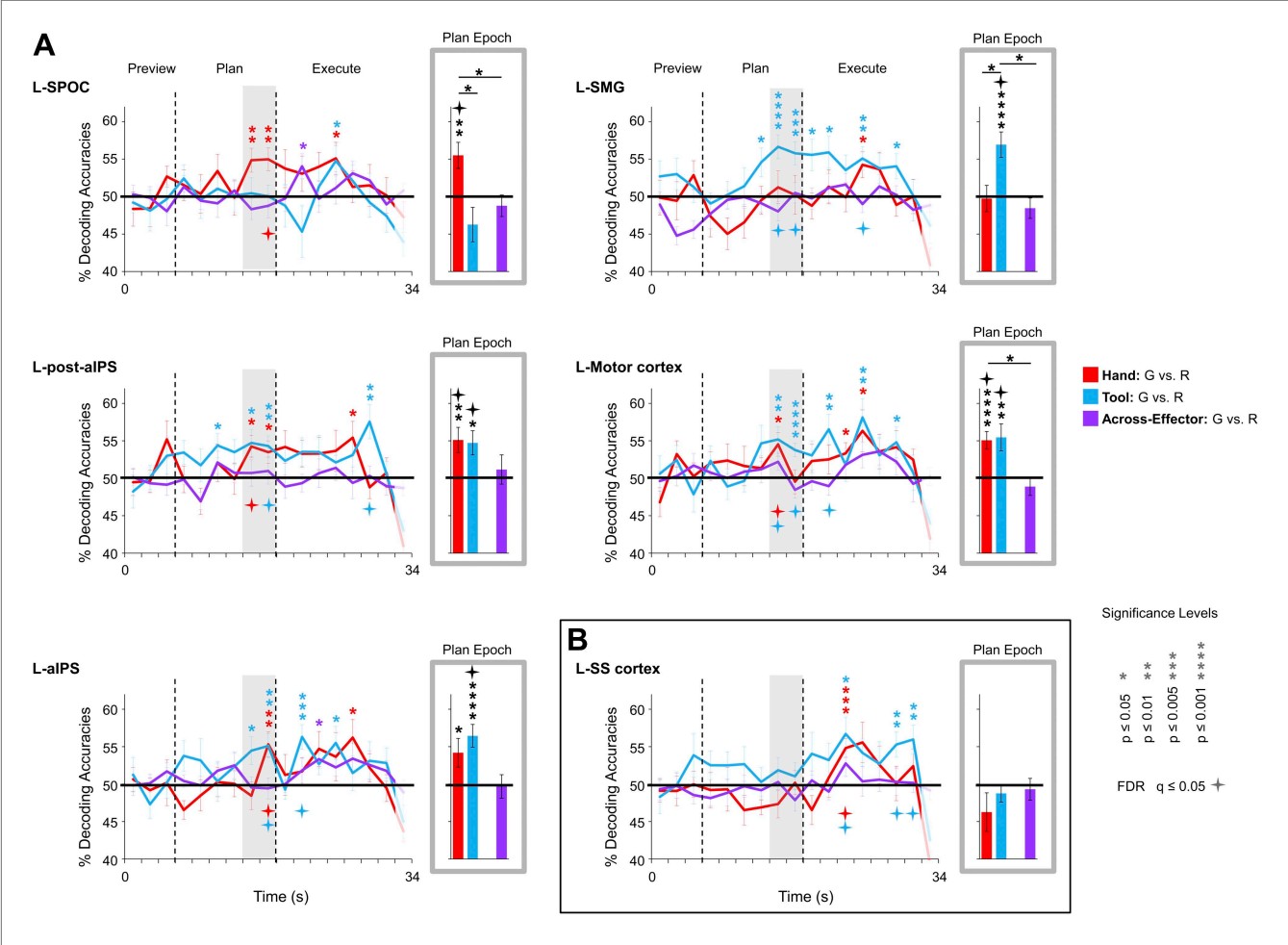

**Figure 3**. Separate movement plans for the hand and tool decoded from frontoparietal cortex. Decoding accuracies are shown for each time point in the trial (time-resolved decoding) and for the Plan-epoch only, the latter based on a windowed average of the spatial activity patterns denoted by the gray shaded bars in the time-resolved decoding plots. In the time-resolved decoding plots, vertical lines correspond to the onset of the Plan and Execute phases of each trial (from left to right). For decoding accuracies discriminating grasp vs reach actions with the Hand (in red) and Tool (in blue) classifier training and testing was done using a single trial N-1 cross-validation approach. Across-effector decoding accuracies (in purple) were computed using all the available data and from training classifiers on Hand-G vs Hand-R trials and testing on Tool-G vs Tool-R trials and then averaging these values with the opposite train-and-test ordering, within each subject. (**A**) Areas of frontoparietal cortex that could decode movement plans with the hand and/or with the tool but not between hand and tool (i.e., no Across-effector decoding). (**B**) Decoding accuracies from the sensory control region, SS-cortex. Note that SS-cortex significantly decodes movements only following action onset (and not during planning). Error bars represent standard error of the mean (SEM) across subjects. Solid black horizontal lines are chance accuracy level (50%). Asterisks assess statistical significance with two-tailed t-tests across subjects with respect to 50%. Four-pointed stars assess statistical significance based on a false discovery rate (FDR) correction of q ≤ 0.05. Note also that in the time-resolved decoding plots, the color of each asterisk/star denotes which specific pair-wise discrimination is significant at each point in time. G: grasp; R: reach.

The following figure supplements are available for figure 3:

**Figure supplement 1**. Classifier decoding accuracies in non-brain control regions.

(*Gallivan et al., 2011a*, *2011b*; *Gallivan et al., 2013*), MVPA in these two areas showed no accurate decoding for any phase of the trial (*Figure 3—figure supplement 1*).

Three general observations can be made based on the results of these decoding analyses. First, predictive movement information, if it is to emerge, generally arises in the two time points prior to initiation of the movement (although note that in a few brain areas, such as L-pIPS and L-PMd, this information is also available prior to these two time points). Second, in support of the notion that this predictive motor information is directly related to the 'intention' to make a movement, accurate

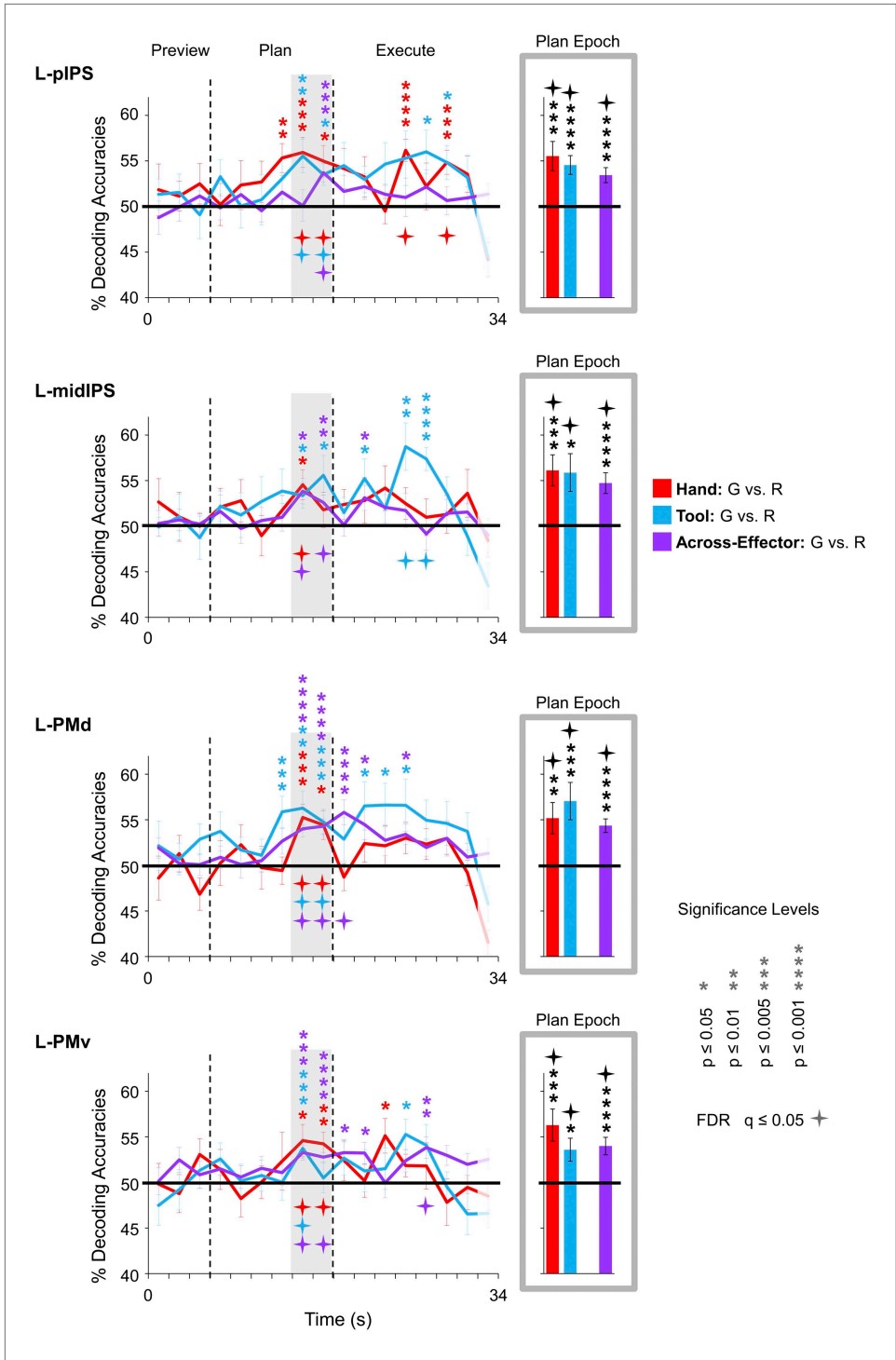

**Figure 4**. Shared movement plans for the hand and tool decoded from frontoparietal cortex. Decoding accuracies are plotted and computed the same as in **Figure 3**. Significant across-effector decoding (purple traces) shows where and when the movement action (Grasp vs Reach) is being represented with some invariance to the acting effector (Hand vs Tool). See **Figure 3** caption for format.

The following figure supplements are available for figure 4:

**Figure supplement 1**. Time-resolved and plan-epoch decoding accuracies for across-effector classification, separated according to the direction of classifier training and testing.

*Figure 4. Continued on next page*

*Figure 4. Continued*

**Figure supplement 2**. Voxel weight analyses for the plan-epoch activity in the cross-decoding ROIs (L-pIPS, L-midIPS, L-PMd, and L-PMv), shown for two representative subjects (in A and B).

**Figure supplement 3**. Movement instructions decoded from transient (but not sustained) responses in auditory cortex.

classification never arises prior to the subject being aware of which action to execute (i.e., prior to the auditory instruction delivered at the initiation of the Plan phase). Finally, decoding related to the planning of a movement can be fully disentangled from decoding related to movement execution, which generally arises several imaging volumes later.

## Separate and shared representations for the hand and tool

Expanding on these MVPA results—and perhaps more important to the overall interpretations of our findings—we next examined in which brain areas the final action (grasping vs reaching) was being represented with some invariance to the effector to be used. To do this, we trained pattern classifiers to discriminate Hand-G vs Hand-R trials and then tested their performance in discriminating Tool-G vs Tool-R trials (the opposite train-and-test process—train set: Tool-G vs Tool-R → test set: Hand-G vs Hand-R—was also performed, and then we averaged the accuracies from both approaches) (for this technique, see also *Formisano et al., 2008*; *Harrison and Tong, 2009*; *Gallivan et al., 2011a*). If successful, this cross-classification would suggest that the object-directed action plans being decoded are to some extent independent of the acting effector (at least to the extent that accurate across-effector classification can be achieved). When we performed this analysis, we found accurate across-effector classification in four regions during planning: two areas in posterior parietal cortex (PPC), pIPS and midIPS, and two areas in premotor cortex, PMd and PMv (see purple decoding traces and bars in *Figure 4*). (Note that separating these tests, Train set: Hand →Test set: Tool and Train set: Tool → Test set: Hand, revealed no major asymmetries in classification, see *Figure 4—figure supplement 1*). Importantly, recall that because the object location was changed (with respect to fixation) between hand and tool experimental runs coupled with the fact that the reverse tool required operating mechanics opposite from those required when the hand was used alone, accurate across-effector classification cannot be attributed to low-level visual, haptic, or kinematic similarities between hand and tool trials. Furthermore, note that accurate across-effector classification does not simply arise in 'any' area where the pattern classifiers are able to successfully discriminate grasp vs reach movements for both the hand and tool. Indeed, although several other areas accurately differentiated the two upcoming movements for both effectors (e.g., post. aIPS, aIPS, t-aIPS, and motor cortex), the preparatory spatial patterns of activity in these areas did not allow for accurate cross-classification. This finding is in itself notable, as it suggests that these latter areas may contain separate coding schemes for the hand and tool. One obvious interpretation of this result is that these latter areas separately code the kinematics used to operate the hand vs tool, providing a neural instantiation of the effector-specific representations thought to be critical for complex tool use. These findings are summarized in *Figure 7*.

We examined whether the qualitative differences in decoding accuracies between the three pairwise comparisons within each region (i.e., within-hand decoding, within-tool decoding and across-effector decoding) reached statistical significance. We reasoned that a brain area involved in coding the hand, for example, might show significantly higher decoding accuracies for actions planned with the hand vs tool. A 13 (number of ROIs) × 3 (number of pairwise comparisons per ROI) repeated-measures ANOVA (rmANOVA) of the plan-epoch decoding accuracies revealed a strong trend towards a significant interaction even in a relatively low-powered omnibus test ($F_{5.701}$ = 2.170, p=0.069, Greenhouse-Geisser [GG] corrected), suggesting differences in the patterns of decoding across regions. Further investigation of the decoding accuracies within each area (using GG-corrected rmANOVAs and False Discovery Rate [FDR]-corrected follow-up paired sample *t*-tests) revealed only a few significant effects: in L-SPOC, decoding accuracies for the hand were significantly higher than for the tool and for across-effector decoding (both at p<0.05; $F_{1.538}$ = 6.084, p=0.014); in L-SMG, decoding accuracies for the tool were significantly higher than for the hand and for across-effector decoding (both at p<0.05; $F_{1.959}$ = 10.016, p<0.001); in L-motor cortex, decoding accuracies for the hand were significantly higher

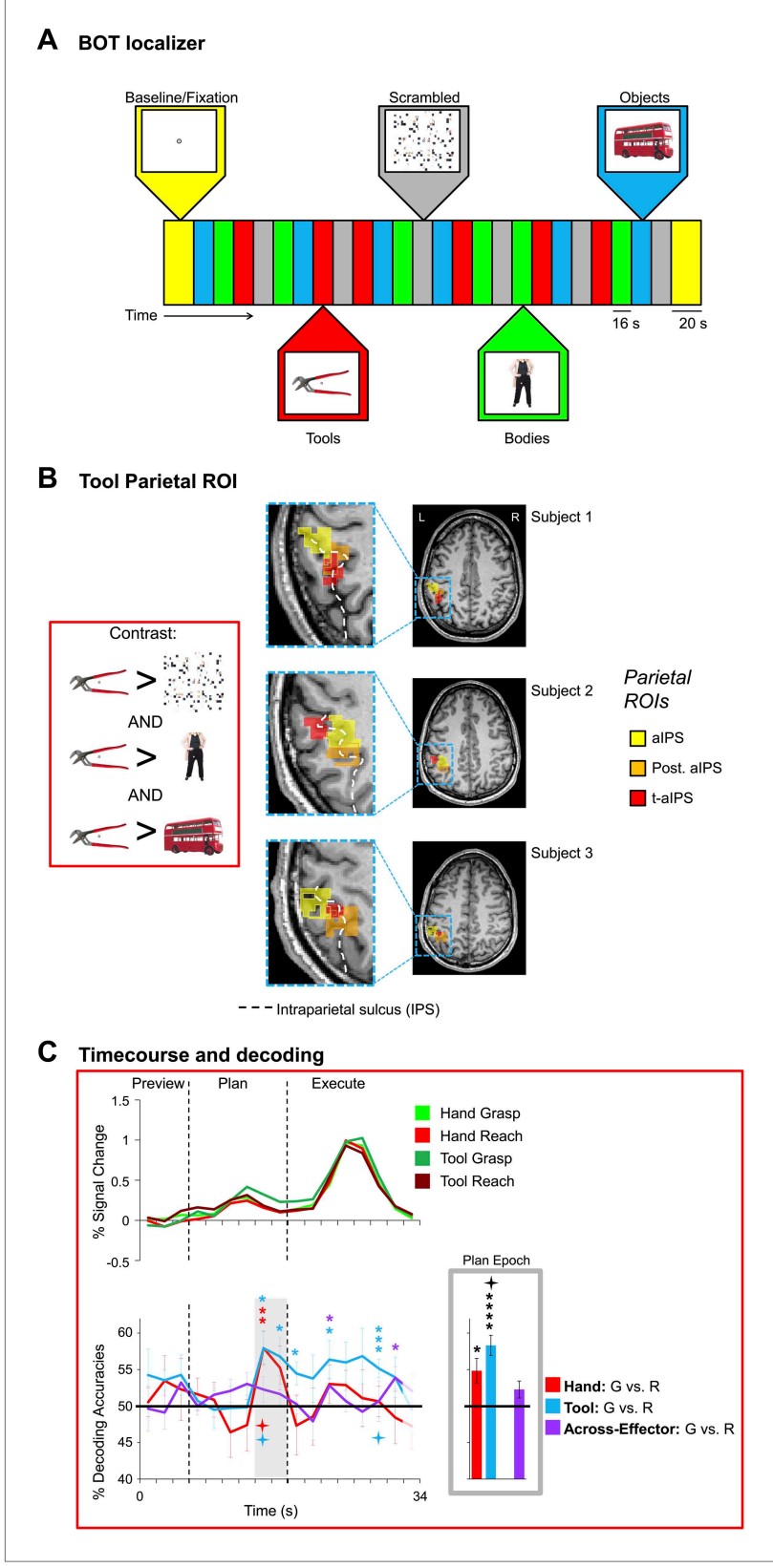

**Figure 5**. Hand and Tool movement plans decoded from the localizer-defined t-aIPS. (**A**) Block-design protocol and experimental timing of the Bodies, Objects, and Tools (BOT) localizer. (**B**) Overlay of tool and anterior parietal ROIs. The Motor experiment-defined anterior parietal ROIs (post. aIPS and aIPS; defined by the

*Figure 5. Continued on next page*

*Figure 5. Continued*

[(Plan + Execute) > 2*(Preview)] contrast) and the Localizer experiment-defined anterior parietal ROI (t-aIPS; defined by the [(Tools > Scrambled) AND (Tools > Bodies) AND (Tools > Objects)] conjunction contrast) are superimposed on the transverse anatomical slices of three representative subjects. Across all subjects we found a reasonable degree of overlap between the Motor and Localizer experiment-defined anterior parietal ROIs. (**C**) % SC time-course activity and time-resolved and plan-epoch decoding accuracies from t-aIPS. See *Figure 3* caption for format.

than across-effector decoding (p<0.05; $F_{1.398}$ = 6.239, p=0.016); and lastly, in L-pMTG, decoding accuracies for the tool were significantly higher than for the hand (p=0.049; $F_{1.968}$ = 4.171, p=0.037) (note that in L-EBA, although decoding accuracies for the hand showed a trend to be higher than for the tool, this did not reach significance; p=0.106; $F_{1.370}$ = 3.635, p=0.078). Taken together, these analyses suggest tool-specific decoding in SMG and pMTG and hand-specific decoding in SPOC and EBA.

## Voxel weight analyses

To further examine the underlying patterns of activity that led to accurate decoding and cross-decoding we investigated the voxel weights assigned by the classifier (where the direction of the weight indicates the relationship of the voxel with the class label, as learned by the classifier; see also the caption for *Figure 4—figure supplement 2*). In particular, we looked for correspondence in the voxel weights across pair-wise comparisons within single subjects as a potential explanation for why the spatial activity patterns in certain areas might show across-effector decoding (the data from two representative subjects is shown in *Figure 4—figure supplement 2*; see also *Formisano et al., 2008* for a similar approach). That is, if the exact same population of voxels were responsible for driving the observed across-effector classification effects than this same voxel set might be consistently biased towards coding one type of action vs the other (i.e., grasping or reaching) for both effectors (hand and tool). (Note that because our pattern classification analysis was performed on non-Talairached data [MVPA was in fact performed on single-subject ACPC-aligned data], comparing the weights across subjects on a single cortical surface was not feasible).

Visual inspection of the voxel weightings failed to reveal any structured or consistent topography within or across subjects (for similar results, see also *Harrison and Tong, 2009*; *Gallivan et al., 2011a*). That is, while the weightings of some voxels appeared to be consistent across pair-wise comparisons within a subject, others appeared to shift their weighting depending on the effector to be used in the movement. (Note that the only consistency observed was that voxels coding for one particular type of action [as indicated by the positive or negative direction of the weight] tended to spatially cluster [which is sensible given the spatial blurring of the hemodynamic response; see *Gallivan et al., 2011a* for a further discussion of this issue]). One possible explanation for the anisotropies observed in the voxel weight distributions across pair-wise comparisons is that they relate to the fact that the decoding accuracies reported here, while statistically significant, are generally quite low (means across participants ~55%). This indicates some appreciable level of noise in the measured planning-related signals, which, given the highly cognitive nature of planning and related processes, likely reflects a wide range of endogenous factors that can vary throughout the course of an entire experiment (e.g., focus, motivation, mood, etc.). Indeed, even when considering the planning-related activity of several frontoparietal structures at the single-neuron level, responses from trial to trial can show considerable variability (e.g., *Snyder et al., 1997*; *Hoshi and Tanji, 2006*). When extrapolating these neurophysiological characteristics to the far coarser spatial resolution measured with fMRI, it is therefore perhaps to be expected that this type of variability should also be reflected in the decoding accuracies generated from single-trial classification. With regards to the resulting voxel weights assigned by the trained SVM pattern classifiers, it should be noted that even in cases where brain decoding is quite robust (e.g., ~80% for orientation gratings in V1–V4), the spatial arrangement of voxel weights still tends to show considerable local variability both within and across subjects (e.g., *Kamitani and Tong, 2005*; *Harrison and Tong, 2009*).

## Control findings in auditory cortex

One alternative explanation to account for the accurate across-effector classification findings reported may be that our frontoparietal cortex results arise not because of the coding of effector-invariant movement goals (grasp vs reach actions) but instead simply because grasp vs reach movements for

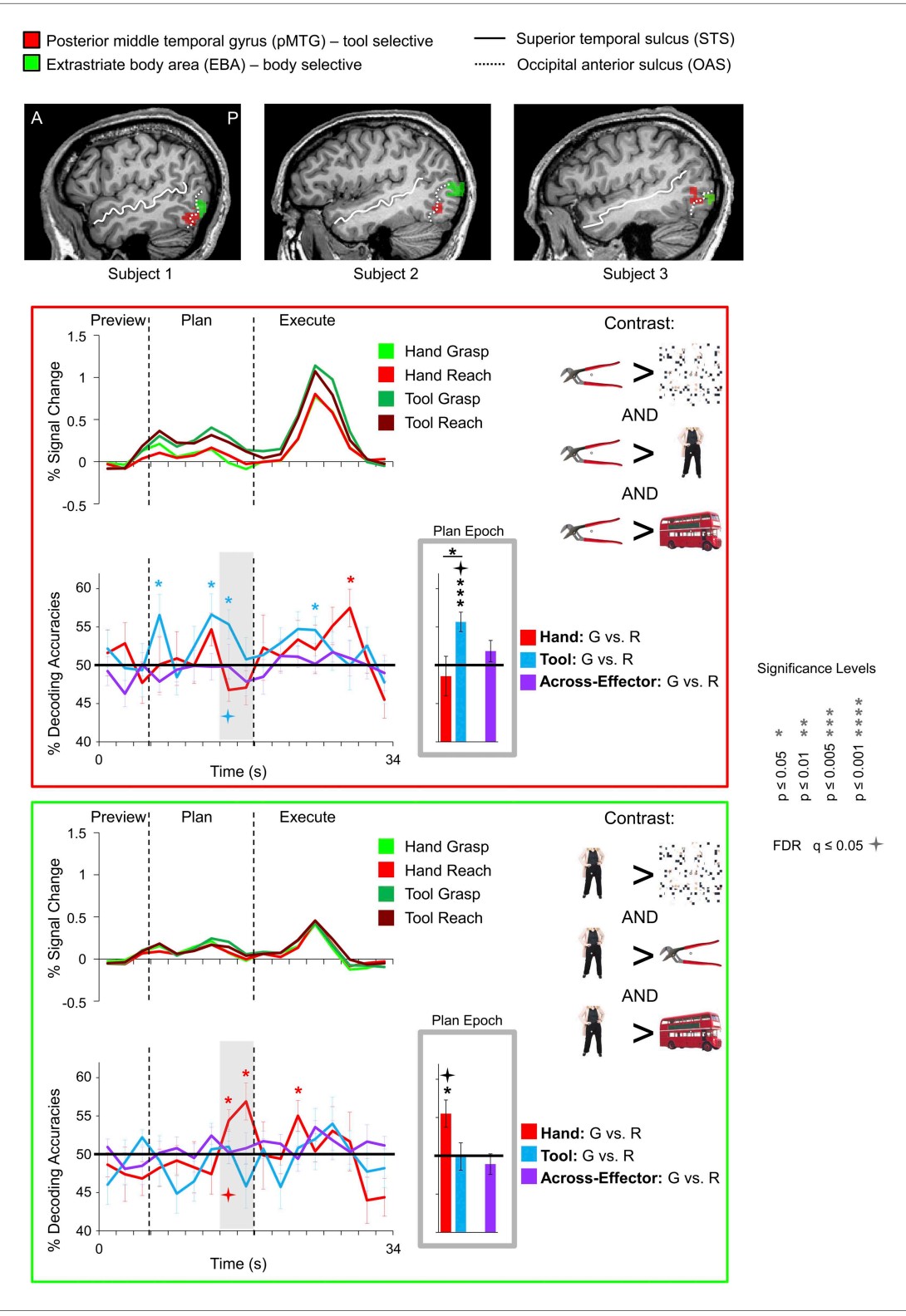

**Figure 6**. Tool and hand movement plans decoded from the localizer-defined pMTG and EBA, respectively. (Top) The pMTG (in red) and EBA (in green) are shown in the same three representative subjects as in **Figure 5**. pMTG was defined using the conjunction contrast of [(Tools > Scrambled) AND (Tools > Bodies) AND (Tools > Objects)] in each subject. EBA was defined using the conjunction contrast of [(Bodies > Scrambled) AND (Bodies > Tools) AND (Bodies > Objects)]. (Below) % SC time-course activity and time-resolved and plan-epoch decoding accuracies shown for pMTG (bordered in red) and EBA (bordered in green). See **Figure 3** caption for format.

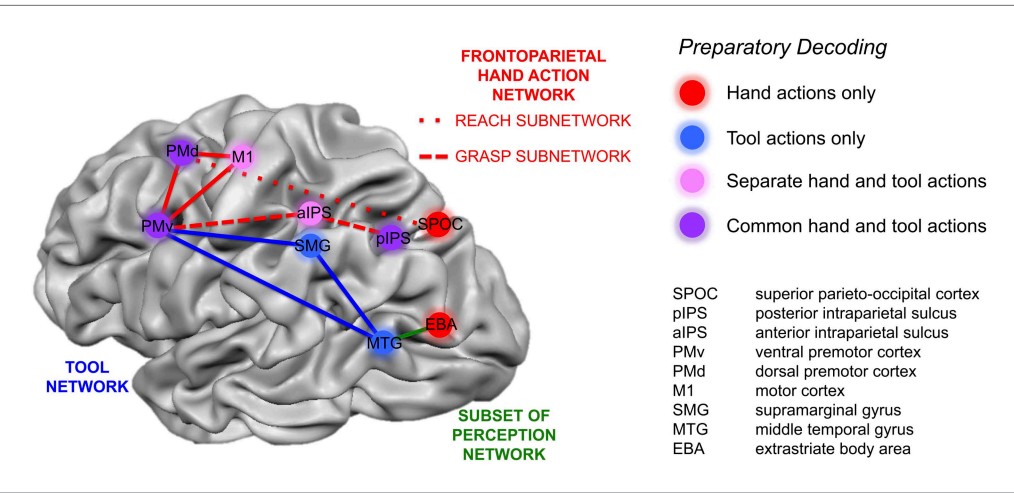

**Figure 7**. Summary of action plan decoding in the human brain for hand and tool movements. Pattern classification revealed a wide range of activity profiles across motor and sensory cortices within networks implicated in hand actions, tool understanding, and perception. Some regions (SPOC and EBA) coded planned actions with the hand but not the tool (areas in red). Some regions (SMG and MTG) coded planned actions with the tool but not the hand (areas in blue). Other regions (aIPS and M1) coded planned actions with both effectors (areas in pink) but did so using different neural representations. A final set of brain areas (pIPS, PMd and PMv) instead coded the final type of action to be performed with invariance as to whether the hand or tool was to be used (areas in purple).

both hand and tool trials are cued according to the same 'Grasp' and 'Reach' auditory instructions. In other words, the cross-decoding observed in PPC and premotor cortex regions might only reflect the selective processing of the auditory commands common to Hand-G and Tool-G ('Grasp') and Hand-R and Tool-R ('Touch') trials and actually have nothing to do with the mutual upcoming goals of the object-directed movement. If this were the case, then we would expect to observe significant across-effector classification in primary auditory cortex (Heschl's gyrus) for the same time-points as that found for PPC (pIPS and midIPS) and premotor (PMd and PMv) cortex. We directly tested for this possibility in our data by separately localizing left Heschl's gyrus in each subject with the same contrast used to define the sensorimotor frontoparietal network, [Plan & Execute > 2*Preview] (recall that auditory cues initiate the onset of the Plan and Execute phases of the trial and so this was a robust contrast for localizing primary auditory cortex). We found that although accurate across-effector classification does indeed arise in Heschl's gyrus during the trial, it does so distinctly earlier in the Plan-phase compared to that of the frontoparietal areas (*Figure 4—figure supplement 3*). This observation is consistent with the noticeably transient percentage signal change response that accompanies the auditory instructions delivered to participants at the beginning of the Plan-phase (see time-course in *Figure 4—figure supplement 3*), as compared to the more sustained planning-related responses that emerge throughout the entire frontoparietal network (*Figure 2*). The temporal disconnect between the cross-decoding found in Heschl's gyrus (which emerges in the fourth volume of the Plan-phase) and frontoparietal cortex (which generally emerges in the fifth-sixth volumes of the Plan-phase) makes it unlikely that the effector-invariant nature of the responses revealed in PPC and premotor cortex can be fully attributable to simple auditory commonalities in the planning cues.

## Limitations of interpretation

It is worth emphasizing that while accurate decoding in a region points to underlying differences in the neural representations associated with different experimental conditions (e.g., for reviews see *Haynes and Rees, 2006*; *Kriegeskorte, 2011*; *Naselaris et al., 2011*; *Norman et al., 2006*), a lack of decoding or 'null effect' (i.e., 50% chance classification) can either reflect that the region 1) is not recruited for the conditions being compared, 2) contains neural/pattern differences between the conditions but which cannot be discriminated by the pattern classification algorithm employed (i.e., a limit of methodology, see *Pereira et al., 2009*; *Pereira and Botvinick, 2011*), or 3) is similarly (but non-discriminately) engaged

in those conditions. With respect to the first possibility, given that we selected frontoparietal cortex ROIs based on their involvement in the motor task at the single-subject level (using the contrast of [Plan & Execute > Preview] across all conditions), it is reasonable to assume that all the localized areas are in some way engaged in movement generation. (Note that this general assumption is confirmed by the higher-than-baseline levels of activity observed in the signal amplitude responses during the Plan- and Execute-phases of the trial in areas of frontoparietal cortex [*Figures 2 and 5*] and that this even appears to be the case in the independently localizer-defined lateral occipitotemporal areas, EBA and pMTG [*Figure 6*]). Although it is understandably difficult to rule out the second possibility (i.e., that voxel pattern differences exist but are not detected with the SVM classifiers), it is worth noting that we do in fact observe null-effects with the classifiers in several regions where they are to be expected. For instance, SS-cortex is widely considered to be a lower-level sensory structure and thus anticipated to only show discrimination related to the motor task once the hand's mechanoreceptors have been stimulated at object contact (either through the hand directly or through the tool, indirectly). Accordingly, here we find that SS-cortex activity only discriminates between grasp vs reach movements following movement onset (i.e., during the Execute phase of the trial). Likewise, in motor cortex we show decoding for upcoming hand- and tool-related actions but, importantly, find no resulting across-effector classification. This latter result is highly consistent with the coding of differences in the hand kinematics required to operate the tool vs hand alone and accords with the presumed role of motor cortex in generating muscle-related activity (*Kalaska, 2009*; *Churchland et al., 2012*; *Lillicrap and Scott, 2013*). These findings in SS-cortex and motor cortex, when combined with the wide-range of decoding profiles found in other areas (i.e., from the hand-selective activity patterns in SPOC and EBA at one extreme, to the tool-selective activity patterns in SMG and pMTG at the other, for summary see *Figure 7*), suggest that the failure of some areas to decode information related to either hand- or tool-related trials (but not those of the other effector) is closely linked to an invariance in the representations of those particular conditions. (To the extent that in cases where the activity of an area fails to discriminate between experimental conditions it can be said that the area is therefore not involved in coding [or invariant to] those particular conditions, we further expand upon interpretations related to these types of null effects in the 'Discussion' section.)

## Discussion

Behavioral, neuropsychological and neurophysiological evidence demonstrates that a central and governing feature of movement planning, and indeed of higher-level cognition in general, is the linking together of overarching action goals with the precise underlying kinematics required by the body to achieve those goals (*Haaland et al., 2000*; *Andersen and Buneo, 2002*; *Fogassi et al., 2005*; *Grafton and Hamilton, 2007*; *Umilta et al., 2008*). Exactly how the human brain supports this cognitive capacity, particularly in the everyday example of tool-use, remains poorly understood. Here we manipulated the type of object-directed hand action that was planned (grasping vs reaching) as well as the effector (hand vs tool) used to implement that action. We then employed fMRI MVPA in order to examine whether planned object-directed hand actions were represented in an effector-specific or effector-independent manner in human frontoparietal and occipitotemporal cortex. At the effector-specific level, we found that SPOC and EBA discriminated upcoming hand movements only whereas SMG and pMTG discriminated upcoming tool movements only. Furthermore, anterior parietal (post. aIPS, aIPS, t-aIPS) and motor cortex areas discriminated planned actions for both the hand and tool, but did not cross-decode between the two effectors. At the effector-independent level, in posterior parietal (pIPS and midIPS) and premotor (PMd and PMv) cortex areas, we found that the pre-movement patterns predictive of grasp vs reach actions for the hand also predicted grasp vs reach actions with the tool. Notably, because the tool-effector required very different hand kinematics than when the hand was used alone, this suggests that these brain areas encoded the action performed rather than the specific muscle movements needed to achieve it. Consistent with the transfer of goals for the hand to those of the tool, this finding resonates with embodied theories of tool use whereby through use, tools become incorporated as part of the body schema. Notably, however, in the majority of regions tested we find that neural representations remain linked to either the hand or tool.

### Representation of the cortical motor hierarchy

Hierarchical theories of motor control have existed for more than a century (*Jackson, 1889*; *Sherrington, 1906*; *Hebb, 1949*), distinguishing between the various levels of abstraction required for action

planning—for example, at the level of muscles, joints, motor kinematics, and movement goals. The present findings provide insights into where different brain regions might be situated within such a hierarchy. For instance, at some lower level along this hierarchy we likely have hand-selective regions like SPOC and EBA and tool-selective regions like SMG and pMTG. Although typically associated with visual-perceptual processing, EBA, like SPOC, has been implicated in coding movements of the hand/arm (*Astafiev et al., 2004*; *Orlov et al., 2010*, although see *Peelen and Downing, 2005*) and the fact that we were unable to decode tool movement plans from these regions suggests that they fail to incorporate tools into the body schema (see also *Gallivan et al., 2009*). SMG and pMTG, in contrast, are typically activated when human subjects view (*Lewis, 2006*; *Peeters et al., 2009*) or pantomime (*Johnson-Frey et al., 2005*) tool-related actions, and damage to these areas creates difficulty in pantomiming or performing tool use actions (*Haaland et al., 2000*). That planning-related signals in SMG and pMTG are able to 'predict' real tool actions, as shown here, provides an important extension of these previous findings, demonstrating that these areas also play an important and selective role in generating object-directed tool actions.

We also found several parietal and frontal brain regions (post. aIPS, aIPS, t-aIPS and motor cortex) that, although able to predict upcoming grasp vs reach movements with both the hand and the tool, did not generalize across the effector (i.e., no across-effector classification). When considering the particular tool used here—where the operating mechanics of the tool were opposite to those of the hand alone—this effector-specific level of action planning is imperative. It provides a coding for the kinematic properties and/or dynamics associated with each effector (*Umilta et al., 2008*; *Jacobs et al., 2010*) as well as the other low-level differences that exist between hand and tool trials (e.g., spatial location of target). These features match the known properties of motor cortex; it provides the largest source of descending motor commands to the spinal neurons that produce hand kinematics (*Porter and Lemon, 1993*) and correspondingly, much of its activity can be accounted for in muscle control terms (see *Kalaska, 2009*; *Churchland et al., 2012*; *Lillicrap and Scott, 2013*). In parietal cortex, aIPS has been strongly implicated in grasp planning and execution (e.g., *Murata et al., 2000*; *Culham et al., 2003*). Notably, it has also been implicated in tool use (*Gallivan et al., 2009*; *Jacobs et al., 2010*), but to date, its precise role in tool-related behaviour has remained unclear. The current findings provide two important clarifications with respect to this previous work. First, the anterior IPS is recruited in the planning of tool actions in addition to those of the hand, suggestive of an important role in preparing actions with both effectors. Second, this pattern of findings on its own does not demonstrate that hand and tool actions rely on the same underlying representations, as previously interpreted (e.g., *Rijntjes et al., 1999*; *Castiello et al., 2000*). Rather, as indicated by our cross-classification findings, the representations may differ, perhaps depending on the specifics of the kinematics or object-effector interactions.

At higher-levels within this hierarchy, we also found several areas (pIPS, midIPS, PMd and PMv) that not only discriminated movement plans for the hand and tool, but moreover, did so using a shared neural code. In the human and macaque monkey, the posterior IPS appears to serve a variety of high-level visual-motor- and cognitive-related functions, such as integrating target- and effector-related information for movement (*Andersen and Buneo, 2002*) and encoding 3D features of objects for hand actions (*Sakata et al., 1998*). One possibility, in line with this previous work, is that effector-independent responses in these areas emerge due to a common coding of object features that are more relevant for grasping than reaching. That is, the same set of object features pertinent for grasping with the hand (object contact points, orientation, distribution of mass, etc.) are pertinent for grasping with the tool and a coding of these features may explain why pattern classifiers trained on hand trials can decode actions performed on tool trials (and vice versa). We also found evidence for these same types of effector-independent representations in premotor areas, PMd and PMv. Each area is engaged in hand actions in both the monkey (*Rizzolatti and Luppino, 2001*; *Raos et al., 2004*, *2006*) and human (*Davare et al., 2006*; *Gallivan et al., 2011*) and their implication in higher-level goal-related processing (*Rizzolatti and Luppino, 2001*; *Cisek et al., 2003*), particularly in the case of tool use with PMv (*Umilta et al., 2008*), strongly resonates with the findings reported here.

## Linking perception and action through tool use

The focus of the present work was to reveal, at the level of the actor, how tool use is planned and implemented in the human brain. In addition to providing insights into how action-centred behavior is cortically represented (discussed above) these findings offer a new lens through which to view findings

reported from previous observation-based fMRI studies. To date, nearly all fMRI studies examining action-centred coding have done so by adopting tasks that require the observation of others' actions (*Lewis, 2006*; *Grafton and Hamilton, 2007*; *Peeters et al., 2009*; *Valyear and Culham, 2010*), in which most commonly, 2D static images or movies of action-related behaviors or tool use are passively viewed by participants (*Lewis, 2006*; *Grafton and Hamilton, 2007*; *Peeters et al., 2009*; *Valyear and Culham, 2010*). Notably, the aim of many of these previous investigations has not necessarily been to reveal how the brain plans and executes different actions per se, but instead, to reveal how the brain understands the goals and intentions of an observed actor. This particular line of research has been primarily motivated by the discovery of 'mirror-neurons' in the monkey (*Rizzolatti et al., 2001*), located in inferior parietal and ventral premotor cortex (*Fogassi et al., 2005*; *Umilta et al., 2008*), which discharge both when the monkey performs a motor act and when the monkey views the same act performed by another individual. When embedded within this larger context, however, it becomes important to not just understand how the actions of other individuals are represented but also how these perceptual representations may relate to the coding of self-generated motor actions. With respect to the latter, past fMRI research has largely left open the question of how goal-directed movements, particularly in the case of tool use, are cortically represented.

Here we provide compelling evidence for a strong coupling between the categorical-selectivity of a brain region, as defined through visual-perceptual processing, and its specific role in behavior, as defined through movement planning. For instance, in occipitotemporal cortex we found that the preparatory activity in the independently localized body-selective EBA and the tool-selective pMTG decoded movement plans for hand and tool actions, respectively. This indicates that, similar to the highly modular nature of visual-perceptual processing in occipitotemporal cortex (*Downing et al., 2006*; *Kanwisher, 2010*), hand- and tool-related actions at certain cortical processing levels may also recruit distinct neural populations. As an interesting departure from these occipitotemporal cortex results, we found that we could decode upcoming movements for 'both' the hand and tool from the independently defined tool-selective t-aIPS. At the functional level, the decoding of tool actions in t-aIPS is entirely congruent with its activation in observation-based tool-related tasks (for reviews, see *Lewis, 2006*; *Frey, 2007*) and, at the anatomical level, the decoding of hand actions in t-aIPS accords with its close proximity to parietal areas involved in hand preshaping and manipulation (*Culham et al., 2003*; *Valyear et al., 2007*; *Gallivan et al., 2011*). When compared to the findings in occipitotemporal cortex, this result indicates that hand and tool movement planning may only begin recruiting similar neural structures at the level of parietal cortex.

The decoding of planned hand- and tool-related actions in EBA and pMTG, respectively, raises important questions as to what exactly is being represented in these two occipitotemporal cortex regions. Although others have shown that hand/arm movements can activate different regions in occipitotemporal cortex (*Astafiev et al., 2004*; *Filimon et al., 2009*; *Cavina-Pratesi et al., 2010*; *Oosterhof et al., 2010*; *Orlov et al., 2010*), here we demonstrate that these signals reflect the 'intention' to perform a motor act rather than the sensory feedback responses (visual, proprioceptive, tactile) that accompany it. This distinction is important because it indicates that these occipitotemporal cortex areas may play a significant role in action planning and control, possibly by predicting the sensory consequences of actions/movement even before those consequences unfold. Given the delay of incoming sensory signals, this type of forward-state estimation is featured prominently in models of action control (*Wolpert and Ghahramani, 2000*; *Wolpert and Flanagan, 2001*) and, from the standpoint of perception, predicting the sensory consequences of movement can be used to disambiguate movements of the body (self) vs movements of the world (others) (*von Helmohltz, 1866*). The current findings would suggest that such forward-state estimations, at least at the level of occipitotemporal cortex, remain linked to the type of effector (hand vs tool) to be used in an upcoming movement. Updating the considerably simpler notion that action planning, particularly in the case of tool use, merely involves 'access' to ventral stream resources (*Milner and Goodale, 1995*; *Valyear and Culham, 2010*), these findings show that hand- and tool-related action plans can actually be decoded from preparatory signals in body- and tool-selective occipitotemporal cortex areas.

In addition to suggesting a role for OTC in visual-motor planning, these findings might also shed light on the organizing principles of the ventral visual stream. Several theories have been proposed to account for the categorical-selectivity of responses throughout OTC (e.g., for faces, scenes, bodies, tools, etc), with the majority arguing that this modular arrangement arises due to similarities/differences in the visual structure of the world and/or how it is experienced (*Kanwisher, 2010*). For

example, according to one prominent view, faces and scenes activate different regions of OTC due to underlying visual field preferences (i.e., faces activate areas with stronger foveal representations, like FFA, whereas scenes activate areas with stronger peripheral representations, like PPA; *Levy et al., 2001*). According to another well-known view, it is instead similarity in visual shape/form that is mapped onto ventral temporal cortex (*Haxby et al., 2001*). One particularly compelling alternative view, however, argues that the organization of OTC may be largely invariant to bottom-up visual properties and that it instead emerges as a by-product of the distinct connectivity patterns of OTC areas with the rest of the brain, particularly the downstream motor structures that use the visual information processed in OTC to plan movements of the body (*Mahon et al., 2007*; *Mahon and Caramazza, 2009*, *2011*). Under this view, the neural specificity frequently observed for the visual presentation of body parts and/or tools in particular regions of OTC may reflect, to a certain extent, their anatomical connectivity with frontoparietal areas involved in generating movements of the body and/or interacting with and manipulating tools, respectively—a notion that garners some empirical support from the 'downstream' functional connectivity patterns of areas involved in body part- and tool-related processing (*Mahon et al., 2007*; *Bracci et al., 2012*). Assuming the sharing of action-related information within functionally interconnected circuits, this conceptual framework might help explain the matching object-selective and planning-related responses observed here within both EBA and pMTG. This compatibility of visual- and motor-related responses within single brain areas resonates with neurophysiological findings in macaque parietal cortex showing that the visual-response selectivity of neurons in AIP (for size, shape, orientation, etc.) are often matched to their motor-response selectivity during action (e.g., *Murata et al., 2000*). This coupling is thought to mediate the transformation of visual information regarding physical object properties into corresponding motor programs for grasping or use (*Jeannerod et al., 1995*; *Rizzolatti and Luppino, 2001*) and resonates with the broader concept of motor affordances, whereby the properties of objects linked to action are automatically represented in movement-related areas of the brain (*Cisek, 2007*; *Cisek and Kalaska, 2010*). Where exactly the current findings fit within the context of these broader frameworks remains unclear, nevertheless, our results provide novel evidence suggesting that the specificity of visual object categorical responses in OTC are in some way linked to a specific role in preparing related motor behaviors.

## Materials and methods

### Subjects

Thirteen right-handed volunteers participated in the Motor experiment (seven females; mean age: 25.7 years, age range: 20–33 years) and were recruited from the University of Western Ontario (London, Ontario, Canada). Eight of these same participants (four females) participated in a second Localizer experiment. All subjects had normal or corrected-to-normal vision and were financially compensated for their participation. Informed consent and consent to publish was obtained in accordance with ethical standards set out by the Declaration of Helsinki (1964) and with procedures approved by the University of Western Ontario's Health Sciences Research Ethics Board (ethics review number: 13507). Subjects were naive with respect to hypothesis testing.

### Motor experiment

#### Setup and apparatus

Each subject's workspace consisted of a black platform placed over the waist and tilted away from the horizontal at an angle (~15°) to maximize comfort and target visibility. To facilitate direct viewing of the workspace, we also tilted the head coil (~20°) and used foam cushions to give an approximate overall head tilt of 30° (*Figure 1A*). Participants planned and performed individual movements with their hand or a tool (reverse tongs) towards a single centrally located object when required (use of the hand and tool were alternated across experimental runs). To minimize limb-related artifacts, participants had the right upper arm braced, limiting movement to the elbow, creating an arc of reachability (*Figure 1B*). The target object was made of white LEGO pieces (length: 7 cm × depth: 3 cm × height: 3 cm) and was secured to the workspace at one of two locations along the arc of reachability for the effector (hand or tool) to be used during each experimental run. The exact placement of the target object for hand and tool trials on the platform was adjusted to match each participant's arm/tool length such that all required movements were comfortable. To mark the object location for hand runs, the target object

was placed within reach by the participant's right hand at a central position on the platform in line with the point of fixation and oriented to maximize the comfort for hand grasping. To mark the object location for tool runs, the target object was placed within reach of the tool by the participant at a further central position, in line with the point of fixation and with the same orientation as that used for the hand. Once marked and prior to initiation of each run type (Hand or Tool), the target object was secured to the platform at one of these two corresponding locations (*Figure 1B*).

During the experiment, the target object was illuminated from the front by a bright white Light Emitting Diode (LED) attached to flexible plastic stalks (Loc-Line; Lockwood Products, Lake Oswego, OR). During participant setup, the illuminator LED was positioned so as to equally illuminate both the hand and tool locations of the target object. Experimental timing and lighting were controlled with in-house software created with MATLAB (The Mathworks, Natick, MA). To control for eye movements, a small green fixation LED was placed above and at a further depth location than the pre-specified tool target position and subjects were required to always foveate the fixation LED during scanning. Eye fixation and arm movements were examined off-line from videos recorded using an MR-compatible infrared-sensitive camera (MRC Systems GmbH, Heidelberg, Germany) positioned underneath the fixation LED and directed toward the subject's eyes and hand.

For each trial, subjects were required to perform one of two actions upon the target object, following a delay period: 1) reach towards and precision grasp (G) the object ('Grasp' auditory command) without lifting and 2) reach towards (R) and manually touch the top of the object ('Touch' auditory command). For experimental runs with the hand, grasping required the subjects to precision grasp the object with their thumb and index finger (Hand-G) without lifting whereas the reaching action required the subject to simply transport their hand to the object without hand pre-shaping (Hand-R). For experimental runs with the tool, grasping required the subjects to precision grasp the object with the set of reverse tongs without lifting, which involved squeezing the handle of the tongs (in order to initially open the distal ends of the tongs) and then subsequently releasing pressure on the handle (in order to close the distal ends of the tongs onto the object; Tool-G, 'Grasp' auditory command). Reaching actions with the reverse tongs simply required the subject to transport the tool to the object without any manipulation, and touch the top of the target (Tool-R; 'Touch' auditory command). Participants were instructed to keep the timing of movements for grasping and reaching trials as similar as possible. Other than the execution of these hand and tool actions, the hand throughout all other phases of the trial (Preview phase, Plan phase and ITI) was to remain still and in a relaxed 'home' position on the right surface of the platform. For each participant the home/starting position was marked with a small elevation of black tape and subjects were required to always return to this same position following execution of the instructed movement. For experimental runs with the hand, the required home position of the hand was a relaxed fist, and for experimental runs with the tool, the required home position was to have the thumb and index finger gently placed on the handle of the tool (without applying pressure). Importantly, within each experimental run, the target object never changed its centrally located position, thus eliminating retinal differences within the workspace of each effector across trials. Critically, however, this manipulation allowed us to maintain large retinal differences (i.e., position of the object with respect to fixation) and somatosensory differences (presence or absence of the tool in hand) between hand and tool runs. Although including hand and tool trials within the same run would have enabled direct statistical comparisons between them, this would have necessitated insertion and removal of the tool during experimental testing, possibly leading to additional movement artifacts. In between runs, the tool was given to and removed from the subject by the experimenter.

We chose a reverse set of tongs as the tool to be used in this experiment because it provided an opposite mapping between the proximal movements of the hand and the distal movements of the tool (i.e., when the hand closed on the reverse tongs, the end of the tongs opened, and vice versa). This incongruence was imperative to the aims of the study (i.e., decoding planned actions independent of the specific muscle activations required) because it allowed the object-directed motor plans for both effectors (hand and tool) to be held constant across the experiment (i.e., grasping or reaching), while at the same time, uncoupling the lower-level hand kinematics required to operate each effector. In contrast, when a normal set of tongs are used, the distal ends of the tool exactly mirror the movements made by the hand (i.e., when the hand closes on the tongs, the distal ends of the tongs would also close), and if we had used this type of tool instead, it would have made it difficult to rule out that any tool-related decoding was independent of the planned hand movements required to operate the tool (See also *Umilta et al., 2008*).

## Experiment design and timing

To extract the visual-motor planning response for the hand and tool from the simple visual and motor execution responses, we used a slow event-related planning paradigm with 34 s trials, each consisting of three distinct phases: 'Preview', 'Plan' and 'Execute' (*Figure 1C*). We adapted this paradigm from previous fMRI work with eye- and arm-movements that have successfully isolated delay period activity from the transient neural responses following the onset of visual input and movement execution (*Curtis et al., 2004*; *Beurze et al., 2007*, *2009*; *Pertzov et al., 2011*) and from other previous studies from our lab in which we successfully used the spatial voxel patterns of delay period responses in order to show that different upcoming movements can be accurately predicted (*Gallivan et al., 2011a*; *2011b*).

In our task, each trial began with the Preview phase, where the subject's workspace was illuminated revealing the centrally located target object. After 6 s of the Preview phase, subjects were given an auditory cue (0.5 s), either 'Grasp' or 'Touch', informing them of the upcoming movement required; this cue marked the onset of the Plan phase. Although there were no visual differences between the Preview and Plan phase portions of the trial (i.e., the single object was always visually present), only in the Plan phase did participants have the necessary motor information in order to prepare the upcoming movement. After 12 s of the Plan phase, a 0.5-s auditory beep cued participants to immediately execute the planned action, initiating the Execute phase of the trial. 2 s following the beginning of this Go cue, the illuminator was turned off, providing the cue for subjects (during both hand and tool runs) to return the hand to its peripheral starting position. After the illuminator was extinguished, subjects then waited in the dark while maintaining fixation for 14 s, allowing the BOLD response to return to baseline prior to the next trial (ITI phase). The two trial types (grasp or reach), with ten repetitions per condition (20 trials total) were randomized within a run and balanced across all runs (that required the same effector) so that each trial type was preceded and followed equally often by every other trial type across the entire experiment.

Separate practice sessions were carried out before the actual experiment to familiarize participants with both the mechanics of the reverse tool and the timing of the paradigm, where in particular, the delay timing required the cued action to be performed only at the beep (Go) cue. These sessions were carried out before the subjects entered the scanner as well as during the anatomical scan (collected at the beginning of every experiment). A testing session for each participant included set-up time (~45 min), 8 functional runs (although two subjects participated in 6 and 10 functional runs, respectively) and 1 anatomical scan, and lasted approximately 3 hr. Throughout the experiment, the subject's fixation and hand movements were monitored using an MR-compatible infrared-sensitive camera optimally positioned directly below the fixation point and facing towards the subject. The videos captured during the experiment were analyzed off-line to verify that the subjects were indeed performing the task as instructed. A more rigorous tracking of the eyes was not performed because our eye-tracking system does not work while the head is tilted due to a partial occlusion from the eyelids.

## Localizer experiment

### Setup and apparatus

Each of the localizer runs included stimulus blocks of color photos projected onto a 2D screen consisting of familiar tools (87 different identities), headless bodies (87 different identities; 44 were females), non-tool objects (87 different identities) and scrambled versions of these same stimuli. Each run lasted 7 min 30 s and was composed of six stimulus epochs per condition, seven scrambled epochs, and two fixation/baseline epochs (20 s) placed at the beginning and end of each run (see *Figure 5* for a protocol of one of the localizer runs). Stimulus epochs were organized into sets of three, separated by scrambled epochs, balanced for epoch history within a single run (photos were repeated across runs). To provide a fixation point, a small black circle was superimposed at the center of each image. Each image subtended 15° of the subject's visual field. Stimuli were organized into separate 16 s epochs, with 18 photos per epoch, presented at a rate of 400 ms per photo with a 490-ms inter-stimulus interval. To encourage that participants maintained attention throughout the localizer scans, subjects performed a one-back task throughout, whereby responses were made, via a right-handed button press, whenever two successive photos were identical. In addition to running this localizer, we also collected a high-resolution anatomical image from each of the participating subjects. All subjects participated in at least three of these localizer runs.

## Motor and Localizer experiments

### MRI acquisition and preprocessing

Subjects were scanned using a 3-Tesla Siemens TIM MAGNETOM Trio MRI scanner. The T1-weighted anatomical image was collected using an ADNI MPRAGE sequence (TR = 2300 ms, TE = 2.98 ms, field of view = 192 mm × 240 mm × 256 mm, matrix size = 192 × 240 × 256, flip angle = 9°, 1-mm isotropic voxels). Functional MRI volumes were collected using a T2*-weighted single-shot gradient-echo echo-planar imaging (EPI) acquisition sequence (time to repetition [TR] = 2000 ms, slice thickness = 3 mm, in-plane resolution = 3 mm × 3 mm, time to echo [TE] = 30 ms, field of view = 240 mm × 240 mm, matrix size = 80 × 80, flip angle = 90°, and acceleration factor [integrated parallel acquisition technologies, iPAT] = 2 with generalized auto-calibrating partially parallel acquisitions [GRAPPA] reconstruction). Each volume comprised 34 contiguous (no gap) oblique slices acquired at a ~30° caudal tilt with respect to the plane of the anterior and posterior commissure (AC-PC), providing near whole brain coverage. In the Motor experiment, we used a combination of imaging coils to achieve a good signal:noise ratio and to enable direct viewing without mirrors or occlusion. Specifically, we placed the posterior half of the 12-channel receive-only head coil (6-channels) beneath the head and tilted it at an angle of ~20° and suspended a 4-channel receive-only flex coil over the forehead. In the Localizer experiment, subjects were scanned using a conventional setup (i.e., stimuli projected onto a 2D screen and viewed with a mirror), with a 32-channel receive-only head coil. Functional data from both the Motor and Localizer experiments were aligned to the high-resolution anatomical collected during the Localizer testing session. We reconstructed the cortical surface from one subject from a high-resolution anatomical image, a procedure that included segmenting the gray and white matter and inflating the surface boundary between them. This inflated cortical surface was used to overlay group activation for figure presentation (for *Figure 2* note that voxel activity was spatially interpolated from 3-mm functional iso-voxels to 1-mm functional iso-voxels). All preprocessing and univariate analyses were performed using Brain Voyager QX version 2.12 (Brain Innovation, Maastricht, the Netherlands).

Following slice scan-time correction, 3D motion correction (such that each volume was aligned to the volume of the functional scan closest in time to the anatomical scan), high-pass temporal filtering (5 cycles per run) and functional-to-anatomical co-registration, functional and anatomical images were rotated such that the axial plane passed through the anterior and posterior commissures (AC-PC space) and then transformed into Talairach space. Other than the sinc interpolation inherent in all transformations, no additional spatial smoothing was performed. Talairach data was only used for group voxelwise Random-effects (RFX) analyses in order to display the pre-defined action-related regions of interest (ROIs). For MVPA, these areas were defined anatomically within each subject's AC-PC data. Given that MVPA discriminates spatial patterns across voxels, we have found it beneficial to select ROIs at the single-subject level using the AC-PC data in lieu of the Talairach data (*Gallivan et al., 2011a*; *2011b*).

For each participant, functional data from each session were screened for motion and/or magnet artifacts by examining the time-course movies and the motion plots created with the motion correction algorithms. None of the runs revealed head motion that exceeded 1 mm translation or 1° rotation. In the Motor experiment, error trials—trials where the participant fumbled with the object (two trials, two participants), performed the incorrect instruction (one trial, one participant), or contaminated the plan phase data by slightly moving their limb or eyes or by performing the action before the 'Go' cue (six trials, three participants)—were identified off-line from the videos recorded during the session and were excluded from analysis by assigning these trials predictors of no interest. This generally low error rate more than likely reflects the fact that subjects were well-trained on the motor task before entering the scanner.

## Motor experiment

### Regions of interest (ROIs)

To localize the specific action-related areas in individual subjects in which to implement MVPA, we used a general linear model (GLM) with predictors created from boxcar functions convolved with the Boynton haemodynamic response function (HRF). For each trial, a boxcar function was aligned to the onset of each phase, with a height dependent upon the duration of each phase: i) 3 volumes for the Preview phase, ii) 6 volumes for the Plan phase, and iii) 1 volume for the Execute phase. The ITI was excluded from the model, therefore all regression coeffcients (betas) were defined relative to the

baseline activity during the ITI. In addition, the time-course for each voxel was converted to percent signal change before applying the RFX-GLM.

To specify our pre-defined ROIs and select voxels for MVPA from each subject we searched for brain areas involved in movement generation. To do this, we contrasted activity for movement planning and execution (collapsed over hand, tool, grasp and touch) vs the simple visual response to object presentation, prior to instruction: (Plan & Execute > Preview)—(Plan [Hand-G + Hand-R + Tool-G + Tool-R] + Execute [Hand-G + Hand-R + Tool-G+ Tool-R] >2*Preview [Hand-G + Hand-R + Tool-G + Tool-R]). The resulting statistical map of all positively active voxels in each subject (t = 3, p<0.005, each subject's activation map was cluster threshold corrected [corrected, p<0.05] so that only voxels passing a minimum cluster size were selected; average minimum cluster size across subjects was 113.5 mm$^3$; for details see 'ROI selection', below) was then used to define 10 different ROIs within the left (contralateral) hemisphere: 1) Superior parieto-occipital cortex (SPOC), 2) posterior intraparietal sulcus (pIPS), 3) middle IPS (midIPS), 4) posterior anterior IPS (post. aIPS), 5) anterior IPS (aIPS), 6) Supramarginal gyrus (SMG), 7) Somatosensory (SS) cortex, 8) Motor cortex, 9) Dorsal premotor (PMd) cortex, and 10) Ventral premotor (PMv) cortex. Nine of these ROIs were selected based on their well-documented involvement in movement planning/generation. We selected SS-cortex to function as a sensory control region (i.e., known to respond to transient stimuli [i.e., sensory events], but not expected to participate in sustained movement planning/intention-related processes). The voxels included in each ROI were selected based on all significant activity within a cube of (15 mm)$^3$ = 3375 mm$^3$ centered on pre-defined anatomical landmarks (see 'ROI selection' below for criteria). These ROI sizes were chosen as it not only allowed the inclusion of a sufficient number of functional voxels for pattern classification (an important consideration), but also ensured that a similar number of voxels were included within each ROI and that the regions could be reliably segregated from adjacent activations (for the average number of functional voxels selected across the 13 subjects, see *Table 1*). Critically, given the orthogonal contrast employed to select these 10 areas (i.e., Plan & Execute > Preview), their activity is not directionally biased to show any preview-, plan- or execute-related pattern differences 'between' any of the experimental conditions.

## ROI selection

Left Superior parieto-occipital cortex (SPOC)

- Defined by selecting voxels located medially and directly anterior to the Parieto-occipital sulcus (POS) (*Gallivan et al., 2009*).

  Left posterior IPS (pIPS)

- Defined by selecting activity at the caudal end of the IPS (*Beurze et al., 2009*).

  Left middle IPS (midIPS)

- Defined by selecting voxels half-way up the length of the IPS, centred on the medial bank (*Gallivan et al., 2011a*, *2011b*), near a characteristic 'knob' landmark observed consistently within each subject.

  Left region located posterior to L-aIPS (L-post. aIPS)

- Defined by selecting activity just posterior to the junction of the IPS and Post central sulcus (PCS), on the medial bank of the IPS (*Culham, 2004*).

  Left aIPS (L-aIPS)

- Defined by selecting voxels located directly at the junction of the IPS and PCS (*Culham et al., 2003*).

  Left Supramarginal gyrus (L-SMG)

- Defined by selecting activity along the supramarginal gyrus, directly lateral to the anterior segment of the IPS (*Lewis, 2006*).

  Left somatosensory cortex (L-SS cortex)

- Defined by selecting voxels medial and anterior to the aIPS, encompassing the Post central gyrus and PCS (*Gallivan et al., 2011a*, *2011b*).

Left Motor cortex

- Defined by selecting voxels around the left 'hand knob' landmark in the Central sulcus (CS) (*Yousry et al., 1997*).

Left Dorsal premotor cortex (PMd)

- Defined by selecting voxels at the junction of the Precentral sulcus (PreCS) and Superior frontal sulcus (SFS) (*Picard and Strick, 2001*).

Left Ventral premotor cortex (PMv)

- Defined by selecting activity inferior and posterior to the junction of the Inferior frontal sulcus (IFS) and PreCS (*Tomassini et al., 2007*).

See *Table 1* for details about ROI coordinates and sizes, and *Figure 2* for representative anatomical locations on one subject's inflated brain.

To provide a control for our cross-decoding analyses in frontoparietal cortex (for motivation, see 'Results'), we examined time-resolved and plan-epoch decoding in the left primary auditory cortex (Heschl's gyrus). Using the same contrast and selection criteria as above, this ROI was neuroanatomically defined in each subject by selecting voxels halfway up along the superior temporal sulcus (STS), on the superior temporal gyrus (between the insular cortex and outer-lateral edge of the superior temporal gyrus; see *Meyer et al., 2010*; *Gallivan et al., 2011b*).

To ensure our decoding accuracies could not result from spurious factors (e.g., task-correlated head or arm movements), we also tested the performance of our classifiers in ROIs outside of our action-related network where no statistically significant classification should be possible. To select these ROIs we further reduced our statistical threshold (after specifying the [Plan & Execute > Preview] network within each subject) down to $t = 0$, $p=1$ and selected all positive activation within 3375 mm$^3$ centered on a consistent point 1) within each subject's right ventricle and 2) at a location situated just outside the skull of the right hemisphere, in the AC-PC plane and directly in line with the posterior commissure.

## Localizer experiment

### Regions of interest (ROI)

To independently localize a specific set of object category-selective ROIs in individual subjects in which to implement MVPA (using the Motor experiment data), we used a GLM with predictor boxcar functions aligned to the onset of each stimulus block for each of the conditions (except for the fixation/baseline epochs) and then convolved the predictors with the Boyton HRF. The time-course for each voxel was converted to percent signal change before applying the GLM.

For each individual, data from the localizer scans was used to identify three distinct ROIs based on previous neuroimaging studies: t-aIPS, pMTG, and EBA. The following same procedure was used to define the three ROIs in each individual: The most significantly active voxel(s), or peak, was first identified based on a particular contrast, statistical thresholds were then set to a determined minimum ($t = 3$, $p<0.005$), and the activity up to 3375 mm$^3$ around the peak was selected. Tool-selective (t-aIPS and pMTG) areas were localized based on a conjunction contrast of ([Tools > Scrambled] AND [Tools > Bodies] AND [Tools > Objects]) and the Body-selective (EBA) area was localized based on a conjunction contrast of ([Bodies > Scrambled] AND [Bodies > Tools] AND [Bodies > Objects]). Note that we define a conjunction contrast as a Boolean AND, such that for any one voxel to be flagged as significant, it must show a significant difference for each of the constituent contrasts. See *Table 1* for details about ROI coordinates and sizes, and *Figures 5 and 6* for representative locations on individual subject's brains.

### Multi-voxel pattern analysis (MVPA)

We used the fine-grained sensitivity afforded by MVPA to not only examine if grasp vs reach movement plans with the hand or tool could be decoded from preparatory brain activity (where little or no signal amplitude differences may exist), but more importantly, because it allowed us to question in what areas the higher-level movement goals of an upcoming action were encoded independent of the lower-level kinematics required to implement them. More specifically, by training a pattern classifier to discriminate grasp vs reach movements with one effector (e.g., hand) and then testing whether that

same classifier can be used to predict the same trial types with the other effector (e.g., tool), we could assess whether the object-directed action being planned (grasping vs reaching) was being represented with some level of invariance to the effector being used to perform the movement (see 'Across-effector classification' below for further details).

## Support vector machine classifiers

MVPA was performed with a combination of in-house software (using Matlab) and the Princeton MVPA Toolbox for Matlab (http://code.google.com/p/princeton-mvpa-toolbox/) using a Support Vector Machines (SVM) binary classifier (libSVM, http://www.csie.ntu.edu.tw/~cjlin/libsvm/). The SVM model used a linear kernel function and default parameters (a fixed regularization parameter C = 1) to compute a hyperplane that best separated the trial responses.

## Inputs to classifier

To prepare inputs for the pattern classifier, the BOLD percent signal change was computed from the time-course at a time point(s) of interest with respect to the time-course at a common baseline, for all voxels in the ROI. This was done in two fashions. The first, extracted percent signal change values for each time point in the trial (time-resolved decoding). The second, extracted the percent signal change values for a windowed-average of the activity for the 4 s (2 imaging volumes; TR = 2) prior to movement (plan-epoch decoding). For both approaches, the baseline window was defined as volume − 1, a time point prior to initiation of each trial and avoiding contamination from responses associated with the previous trial. For the plan-epoch approach—the time points of critical interest in order to examine whether we could predict upcoming movements (*Gallivan et al., 2011a*, *2011b*)— we extracted the average pattern across imaging volumes 8–9 (the final 2 volumes of the Plan phase), corresponding to the sustained activity of the planning response prior to movement (*Figures 1D and 2*). Following the extraction of each trial's percent signal change, these values were rescaled between −1 and +1 across all trials for each individual voxel within an ROI. Importantly, through the application of both time-dependent approaches, in addition to revealing which types of movements could be decoded, we could also examine specifically when in time predictive information pertaining to specific actions arose.

## Pair-wise discriminations

SVMs are designed for classifying differences between two stimuli and LibSVM (the SVM package implemented here) uses the so-called 'one-against-one method' for each pair-wise discrimination (*Hsu and Lin, 2002*). We capitalized on this fact in order to characterize different brain regions according to the types of upcoming movements they could predict: Hand-G vs Hand-R and/or Tool-G vs Tool-R. We did not examine individual pair-wise discriminations for movements between hand and tool trials (e.g., Hand-G vs Tool-G, Hand-R vs Tool-R) given the fact that in addition to the differences in action planning, substantial visual and somatosensory differences already exist between the two types of trials, like the retinal position of the target object and presence/absence of the tool which both varied between experimental runs. Notably, while these low-level visual and somatosensory differences between experimental runs provided an inherent impediment for interpreting any direct comparisons between hand and tool trials, the presence of these differences significantly aided the interpretation of accurate across-effector classification results. That is, cross-decoding between effectors for the planned action (grasp vs reach) would be unequivocally independent of any visual or somatosensory similarities between the hand and tool runs.

## Single-trial classification

For each subject and for each of the ten action-related ROIs in the Motor experiment and three perception-related ROIs in the Localizer experiment, separate binary SVM classifiers were estimated for MVPA (i.e., for each pair-wise comparison, Hand-G vs Hand-R and Tool-G vs Tool-R, and for each time point). We used a 'leave-one-trial-pair-out' N-fold cross-validation to test the accuracy of the SVM classifiers (i.e., one trial from each of the conditions being compared [two trials total] were reserved for testing the classifier and the remaining [N − 1] trial pairs were used for classifier training). We performed this N − 1 cross-validation procedure until all trial pairs were tested, and then averaged across N-iterations in order to produce a classification accuracy measure for each pair-wise discrimination and subject (*Duda et al., 2001*). We statistically assessed decoding significance with a two-tailed *t*-test vs 50% chance decoding. To control for the problem of multiple comparisons, a false discovery rate (FDR)

correction of q ≤ 0.05 was applied based on all *t*-tests performed at each time point within an ROI (*Benjamini and Yekutieli, 2001*).

Note that the data being used at any single time point (e.g., each TR in the time-resolved decoding approach) are independent as they are full trial-lengths removed from directly adjacent trials (recall that each trial = 34 s), providing more than adequate time for the hemodynamic responses associated with individual TRs used for classifier testing to sufficiently uncouple (this would not necessarily be the case in a rapid event-related design). Furthermore, the trial orders were fully randomized and so any possible correlations between train and test data is not obvious and should not bias the data towards correct vs incorrect classification (*Misaki et al., 2010*).

### Permutation tests

In addition to the *t*-test, we separately assessed statistical significance with non-parametric randomization tests for the plan-epoch decoding accuracies (*Golland and Fischl, 2003*; *Etzel et al., 2008*; *Smith and Muckli, 2010*; *Chen et al., 2011*; *Gallivan et al., 2011a*, *2011b*). For specific details pertaining to this test see our recent work (*Gallivan et al., 2011a*, *2011b*, *2013*). In sum, the important finding highlighted from these permutation tests is that the brain areas showing significant decoding with the one sample parametric *t*-tests (vs. 50% at p<0.05) 'also' show significant decoding (albeit higher, p<0.001) with the empirical non-parametric permutation tests.

### Across-effector classification

In order to test whether a SVM pattern classifier trained to discriminate between grasp vs reach trial types with one effector could then be used to accurately decode pattern differences when tested with trials belonging to the other effector (e.g., train set: Hand-G vs Hand-R, test set: Tool-G vs Tool-R), instead of using the N − 1 cross-validation procedure (implemented above) we used all the available single trial data for both classifier training and testing (i.e., one train-and-test iteration; *Smith and Muckli, 2010*; *Gallivan et al., 2011a*). Cross-decoding accuracies for each subject were computed by averaging together the two accuracies generated by using each pair of effector-specific trial types for classifier training and testing (e.g., Hand trials were used to train the classifier in one analysis when Tool trials were used for testing, and then they were used to test the classifier in the other analysis when the Tool trials were used for classifier training). The means across subjects of this cross-decoding procedure are reported in *Figures 3–6*. We statistically assessed decoding significance with a two-tailed *t*-test vs 50% chance decoding. A FDR correction of q ≤ 0.05 was applied based on all *t*-tests performed at each time point.

## Acknowledgements

The authors are grateful to Craig Chapman, Fraser Smith, and Melvyn Goodale for helpful discussions and/or comments on previous versions of the manuscript.

## Additional information

### Competing interests

JCC: Reviewing editor, *eLife*. The other authors declare that no competing interests exist.

### Funding

| Funder | Grant reference number | Author |
| --- | --- | --- |
| Canadian Institute of Health Research | MOP84293 | Jody C Culham |
| Banting Fellowship | | Jason P Gallivan |
| Natural Sciences and Engineering Research Council Postdoctral fellowship award | | Kenneth F Valyear |
| Ontario Ministry of Research and Innovation Postdoctral fellowship award | | Jason P Gallivan |

The funders had no role in study design, data collection and interpretation, or the decision to submit the work for publication.

## Author contributions
JPG, Conception and design, Acquisition of data, Analysis and interpretation of data, Drafting or revising the article, Contributed unpublished essential data or reagents; DAM, Conception and design, Acquisition of data, Contributed unpublished essential data or reagents; KFV, JCC, Conception and design, Analysis and interpretation of data, Drafting or revising the article

## Ethics
Human subjects: All subjects had normal or corrected-to-normal vision and were financially compensated for their participation. Informed consent and consent to publish was obtained in accordance with ethical standards set out by the Declaration of Helsinki (1964) and with procedures approved by the University of Western Ontario's Health Sciences Research Ethics Board (ethics review number: 13507). Subjects were naive with respect to hypothesis testing.

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
