## [Decision Letter]

Thank you for choosing to send your work entitled “Decoding the neural mechanisms of human tool use” for consideration at *eLife*. Your article has been evaluated by a Senior editor and 3 reviewers, one of whom is a member of our Board of Reviewing Editors.

The following individuals responsible for the peer review of your submission want to reveal their identity: Dora Angelaki (Reviewing editor) and Brad Mahon (peer reviewer).

The Reviewing editor and the other reviewers discussed their comments before we reached this decision, and the Reviewing editor has assembled the following comments to help you prepare a revised submission.

The paper is clearly written and organized, and tackles an important question with a well-thought-out and sophisticated approach. Overall this is excellent work, although there are some concerns/questions that the authors should address:

1) An aspect of the Results that is not sufficiently brought out up front in the article (Abstract and Introduction) is that in different regions it is possible to distinguish different factors within the experiment. For instance, the fact that in pMTG upcoming actions can be decoded only when they are performed with the tools, whereas in Superior parietal-occipital cortex decoding was higher for the hand than for the tool, is a really important aspect of the findings that is somewhat buried. The authors should bring this out more fully.

2) It seems somewhat surprising that the regions able to decode across effector (tool→hand, hand→tool) are in IPS and premotor cortex, while regions within IPS/premotor cortex were also shown to have better decoding for hand than for tools. How can the authors reconcile these different effects (are they different populations of voxels driving the decoding accuracy?). Perhaps plotting the weights, voxel-wise, could help to address this.

3) It would be good to see the results (e.g., in figure form, and reduced statistics) for the control analysis of decoding of the ventricle and outside of the brain. This analysis addresses a really insidious issue that will be at the forefront of skeptical readers’ minds; it should be dealt with in full (rather than just saying the analysis was performed and there were no effects).

4) In discussing the occipital-temporal decoding of upcoming actions, the authors come close to, but shy away from, considering whether the organization of the ventral stream may be constrained by the way that the ventral stream is connected up with the rest of the brain (e.g., specificity for the visual representations of body parts could be driven by connectivity to regions of somatosensory or motor areas, and the same for regions that represent tools). The authors’ findings, showing that occipital temporal regions can decode upcoming actions would seem to resonate with that type of an explanation of the causes of neural specificity for different classes of visual stimuli in the ventral stream, at least as it applies to the representation of tools.

5) The auditory cue to determine the subjects’ movement seems to have been the same for both the hand and tool conditions. This consistency across effectors could be responsible for some or even all of the apparently cross-effector classification. That is, the activity in the ROIs that show above-chance cross-effector decoding of actions could be due to the content of the cue rather than preparation for the action per se. This issue must be addressed.

6) It was not clear that the authors balanced the number of voxels across ROIs. This is potentially important as other studies have shown that MVPA performance is strongly influenced by the number of voxels tested.

7) The authors averaged across both directions of the cross-effector test (hand→tool and tool→hand). It is possible however that there are asymmetries in action coding that would be revealed by disentangling these tests.

8) While the authors find that the univariate response properties of regions generally correspond to decoding performance with MVPA, this could partly reflect the ROI approach that was used. A whole-brain MVPA mapping approach might well reveal other regions that code action preparation that are not activated (or not selectively activated) in the univariate sense.

9) The authors note the increased sensitivity of their method relative to a univariate approach, but MVPA (like any other method) carries assumptions and limitations that should shape how the data are interpreted. This could be discussed.

10) The authors report MVPA analyses at each time point, yet we did not find much in the discussion of the results about how these decoding effects develop over time.

---

## [Author Response]

*1) An aspect of the Results that is not sufficiently brought out up front in the article (Abstract and Introduction) is that in different regions it is possible to distinguish different factors within the experiment. For instance, the fact that in pMTG upcoming actions can be decoded only when they are performed with the tools, whereas in Superior parietal-occipital cortex decoding was higher for the hand than for the tool, is a really important aspect of the findings that is somewhat buried. The authors should bring this out more fully*.

We thank the reviewers for this comment. Like the reviewers, we also believe that these hand-specific and tool-specific decoding results are a very important aspect of the findings, and, given the wide-range of findings observed and discussed, it is possible that these important results may have appeared somewhat buried in the Discussion section of our initial manuscript. In line with the reviewers’ suggestions we have modified our Abstract and Introduction to help bring these findings saliently to the forefront of our article.

Also, to more fully highlight our effector-specific decoding effects, we now end our Introduction section with the following new paragraph:

“Consistent with an effector-specific coding of hand- and tool-related movements we found that preparatory signals in SPOC and EBA differentiated upcoming movements of the hand only (i.e., hand-specific) whereas in SMG and pMTG they discriminated upcoming movements of the tool only (i.e., tool-specific). In addition, in anterior parietal regions (e.g., aIPS) and motor cortex we found that pre-movement activity patterns discriminated planned actions of *both* the hand and tool but, importantly, could not be used to predict upcoming actions of the other effector. Instead, we found that this effector-independent type of coding was constrained to the preparatory signals of a subset of frontoparietal areas (posterior IPS and premotor cortex), suggesting that in these regions neural representations are more tightly linked to the goal of the action (grasping versus reaching) rather than the specific hand movements required to implement those goals.”

*2) It seems somewhat surprising that the regions able to decode across effector (tool→hand, hand→tool) are in IPS and premotor cortex, while regions within IPS/premotor cortex were also shown to have better decoding for hand than for tools. How can the authors reconcile these different effects (are they different populations of voxels driving the decoding accuracy?)? Perhaps plotting the weights, voxel-wise, could help to address this*.

We expect that this comment from the reviewers directly refers to the time-resolved decoding accuracies within the across-effector decoding regions (pIPS, midIPS, PMd, and PMv, Figure 4) rather than the Plan-epoch decoding accuracies, as the latter reveals very similar levels of decoding for the hand and tool conditions. More specifically, this reviewer comment likely pertains to the time-resolved decoding in L-midIPS (see Figure 4, second plot from top, gray shaded bar) – the *only* region in which decoding for the hand and across-effector conditions are *both* significantly above chance during the final two volumes of the plan phase (at FDR correction levels) and the *tool condition is not*. We fully expect that this observation, more than anything else, speaks to the noisier (and thus more variable) nature of the single time-point decoding approach (in which pattern decoding is performed at each individual imaging volume in the trial), compared to that when time-points are averaged together, producing a mean activity pattern. Indeed, this difference in statistical significance between both approaches (i.e., time-resolved vs. mean-based decoding analyses) has been well-documented and previously reported (e.g., Harrison and Tong, 2009). For this very reason, we also provided (for the benefit of readers as well as reviewers) the ‘less-noisy’ Plan-Epoch-based decoding analysis, in which decoding is performed based on the ‘average’ pattern of voxels over the two imaging volumes preceding movement onset (denoted by the gray shaded bars in Figure 4, for example).

Consistent with previously published findings (Harrison and Tong, 2009), this latter mean-pattern-based approach typically provides a more robust and statistically significant measure of decoding accuracy. However, we should note that despite tool-specific decoding in L-midIPS not passing the FDR correction level in the time-resolved decoding analysis, it is still significant at p<0.05 using the two-sample parametric t-test (vs. 50%). With respect to this last point, it is worth recognizing that we did in fact perform a statistical analysis (included in the main manuscript with our previous submission) comparing decoding accuracies in the mean-based plan-epoch across pairwise comparisons within each region. While the results of these tests revealed several significant effects (e.g., the responses in L-SPOC were found to be hand-specific whereas the responses in L-SMG were found to be tool-specific), we did not find significant differences between decoding accuracies for the hand versus tool conditions in L-midIPS. This finding reinforces the notion that *differences in significance* for hand- and tool-related decoding (vs. 50% chance) do not necessarily equate to *significant differences* between those conditions.

With regards to the reviewer comment: “How can the authors reconcile these different effects (are they different populations of voxels driving the decoding accuracy?)? Perhaps plotting the weights, voxel-wise, could help to address this.” We thank them for providing this suggestion. It is indeed possible that by plotting the voxel weights we might potentially reveal whether it is the same (or different) population of voxels across hand- and tool-selective comparisons that are responsible for driving the observed across-effector classification effects. We have included such an analysis in the revised submission for two representative subjects and also include a detailed description of these results and the analyses employed (see Figure 4—figure supplement 2 and the subheading entitled ‘Voxel weights analyses’). To briefly summarize the findings here, we found, like several previously published interpretations of the voxel weights (e.g., Kamitani and Tong, 2005; Harrison and Tong, 2009; Gallivan et al., 2011a; Gallivan et al., 2011b), that there can be quite a variable topography in the voxel weightings assigned by the classifier within subjects and across pair-wise comparisons, even in the regions that show successful cross-decoding. This spatial variability makes pinpointing the specific voxels responsible for driving the observed cross-classification effects largely speculative. [Note that we have opted to place these additional analyses as a figure supplement as we strongly believe they do not significantly enhance the narrative of the main manuscript.]

*3) It would be good to see the results (e.g., in figure form, and reduced statistics) for the control analysis of decoding of the ventricle and outside of the brain. This analysis addresses a really insidious issue that will be at the forefront of skeptical readers’ minds; it should be dealt with in full (rather than just saying the analysis was performed and there were no effects)*.

We completely agree with this comment from the reviewers that inclusion of control analyses showing decoding from the ventricle and outside of the brain would be of particular benefit to ease sceptical readers’ minds concerning the validity of the effects being reported. We have now included these control analyses (see Figure 3—figure supplement 1) for interested readers. Again, we have opted to include these analyses in a figure supplement rather than within the main manuscript as we believe they do not enhance the narrative.

*4) In discussing the occipital-temporal decoding of upcoming actions, the authors come close to, but shy away from, considering whether the organization of the ventral stream may be constrained by the way that the ventral stream is connected up with the rest of the brain (e.g., specificity for the visual representations of body parts could be driven by connectivity to regions of somatosensory or motor areas, and the same for regions that represent tools). The authors’ findings, showing that occipital temporal regions can decode upcoming actions would seem to resonate with that type of an explanation of the causes of neural specificity for different classes of visual stimuli in the ventral stream, at least as it applies to the representation of tools*.

This is an excellent comment from the reviewers and indeed, we admit that we did somewhat “shy away” from fully discussing this issue in our initiation submission. In drafting our initial manuscript we were uncertain as to how potential reviewers might respond to this type of “speculative” discussion (thus our reason for toning down our arguments in that regard) but we are quite happy, given this encouragement from the reviewers, to now fully engage this possible explanation of our results in the revised manuscript. Moreover, we believe that the capacity to discuss these intriguing possibilities more fully in the revised paper also allows us to highlight the reviewers’ suggestions raised in comment #1.

At the end of our Discussion section, we now close with the following paragraph:

“In addition to suggesting a role for OTC in visual-motor planning, these findings might also shed light on the organizing principles of the ventral visual stream. Several theories have been proposed to account for the categorical-selectivity of responses throughout OTC (e.g., for faces, scenes, bodies, tools, etc.), with the majority arguing that this modular arrangement arises due to similarities/differences in the visual structure of the world and/or how it is experienced (Kanwisher, 2010). For example, according to one prominent view, faces and scenes activate different regions of OTC due to underlying visual field preferences (i.e., faces activate areas with stronger foveal representations, like FFA, whereas scenes activate areas with stronger peripheral representations, like PPA, (Levy et al., 2001)). According to another well-known view, it is instead similarity in visual shape/form that is mapped onto ventral temporal cortex (Haxby et al., 2001). One particularly compelling alternative view, however, argues that the organization of OTC may be largely invariant to bottom-up visual properties and that it instead emerges as a by-product of the distinct connectivity patterns of OTC areas with the rest of the brain, particularly the downstream motor structures that use the visual information processed in OTC to plan movements of the body (Mahon et al., 2007; Mahon and Caramazza, 2009, 2011). Under this view, the neural specificity frequently observed for the visual presentation of body parts and/or tools in particular regions of OTC may reflect, to a certain extent, their anatomical connectivity with frontoparietal areas involved in generating movements of the body and/or interacting with and manipulating tools, respectively – a notion that garners some empirical support from the ‘downstream’ functional connectivity patterns of areas involved in body part- and tool-related processing (Mahon et al., 2007; Bracci et al., 2012). Assuming the sharing of action-related information within functionally interconnected circuits, this conceptual framework might help explain the matching object-selective and planning-related responses observed here within both EBA and pMTG. This compatibility of visual- and motor-related responses within single brain areas resonates with neurophysiological findings in macaque parietal cortex showing that the visual-response selectivity of neurons in AIP (for size, shape, orientation, etc) are often matched to their motor-response selectivity during action (e.g., Murata et al., 2000). This coupling is thought to mediate the transformation of visual information regarding physical object properties into corresponding motor programs for grasping or use (Jeannerod et al., 1995; Rizzolatti and Luppino, 2001) and resonates with the broader concept of motor affordances, whereby the properties of objects linked to action are automatically represented in movement-related areas of the brain (Cisek, 2007; Cisek and Kalaska, 2010). Where exactly the current findings fit within the context of these broader frameworks remains unclear, nevertheless, our results provide novel evidence suggesting that the specificity of visual object categorical responses in OTC are in some way linked to a specific role in preparing related motor behaviors.”

*5) The auditory cue to determine the subjects’ movement seems to have been the same for both the hand and tool conditions. This consistency across effectors could be responsible for some or even all of the apparently cross-effector classification. That is, the activity in the ROIs that show above-chance cross-effector decoding of actions could be due to the content of the cue rather than preparation for the action per se. This issue must be addressed*.

This is a clever comment by the reviewers. It is indeed correct that the auditory cue for both hand and tool conditions was the same (i.e., “Grasp” or “Touch”) and so it is indeed a possibility that this consistency across the effectors could, to some extent, account for the across-effector classification effects reported. We have directly addressed this issue in our revised paper by separately localizing primary auditory cortex activity (i.e., Heschl’s gyrus) in all participants, extracting the activity related to the motor task, and then determining whether we could decode, using the time-resolved and plan-epoch based analyses, the content of the cue (i.e., “Grasp” or “Reach”) from the area.

We have included the results of this analysis (see Figure 4—figure supplement 3). Also, we now write:

“One alternative explanation to account for the accurate across-effector classification findings reported may be that our frontoparietal cortex results arise not because of the coding of effector-invariant movement goals (grasp vs. reach actions) but instead simply because grasp vs. reach movements for both hand and tool trials are cued according to the same “Grasp” and “Reach” auditory instructions. In other words, the cross-decoding observed in PPC and premotor cortex regions might only reflect the selective processing of the auditory commands common to Hand-G and Tool-G (“Grasp”) and Hand-R and Tool-R (“Reach”) trials and actually have nothing to do with the mutual upcoming goals of the object-directed movement. If this were the case, then we would expect to observe significant across-effector classification in primary auditory cortex (Heschl’s gyrus) for the same time-points as that found for PPC (pIPS and midIPS) and premotor (PMd and PMv) cortex. We directly tested for this possibility in our data by separately localizing left-Heschl’s gyrus in each subject for the same contrast used to define the sensorimotor frontoparietal network, [Plan & Execute > 2*Preview](recall that auditory cues initiate the onset of the Plan and Execute phases of the trial and so this was a robust contrast for localizing primary auditory cortex). We found that although accurate across-effector classification does indeed arise in Heschl’s gyrus during the trial, it does so distinctly earlier in the Plan-phase compared to that of the frontoparietal areas (see Figure 4—figure supplement 3). This observation is consistent with the noticeably transient percentage signal change response that accompanies the auditory instructions delivered to participants at the beginning of the Plan-phase (see time-course in Figure 4—figure supplement 3), as compared to the more sustained planning-related responses that emerge throughout the entire frontoparietal network (see Figure 2). The temporal disconnect between the cross-decoding found in Heschl’s gyrus (which emerges in the fourth volume of the Plan-phase) and frontoparietal cortex (which generally emerges in the fifth-sixth volumes of the Plan-phase) makes it unlikely that the effector-invariant nature of the responses revealed in the wide-range of frontoparietal areas can be fully attributable to simple auditory commonalities in the planning cues.”

In addition, in our Materials and methods, we now also include details concerning the localization of auditory cortex. We write:

“To provide a control for our cross-decoding analyses in frontoparietal cortex (for motivation, see *Results*), we examined time-resolved and plan-epoch decoding in the left primary auditory cortex (Heschl’s gyrus). Using the same contrast and selection criteria as above, this ROI was neuroanatomically defined in each subject by selecting voxels halfway up along the superior temporal sulcus (STS), on the superior temporal gyrus (between the insular cortex and outer-lateral edge of the superior temporal gyrus).”

We are very thankful to the reviewers for bringing this important consideration with regards to interpreting the results to our attention.

*6) It was not clear that the authors balanced the number of voxels across ROIs. This is potentially important as other studies have shown that MVPA performance is strongly influenced by the number of voxels tested*.

Our voxel selection criterion for ROIs was constrained by brain anatomy and the statistical significance of the voxels for the contrast of interest (for the frontoparietal ROIs, the contrast selected for voxels involved in movement generation; Plan & Execute > Preview; for the OTC ROIs, the contrast selected for voxels with a maximal response to preferred object category). Specifically, all frontoparietal ROI clusters were selected according to anatomical landmarks (see the section in the Materials and methods entitled “ROI selection”) at the single-subject level at a statistical threshold of t=3, p<0.005 (cluster threshold corrected at p<0.05) using a maximum of (15 mm)^3^ cluster sizes. The localizer-defined ROI clusters were selected using well-documented ROI selection methods for the ventral visual stream (i.e., we selected (15 mm)^3^ of activity around the peak voxel of activity for each contrast of interest). Together, these voxel selection criterions not only ensured that the voxels being selected for each ROI correspond with the selection criteria applied in our previously published work in frontoparietal cortex (Gallivan et al., 2011a; Gallivan et al., 2011b; Gallivan et al., 2013) and others’ work on the ventral stream in EBA and pMTG (e.g., Downing et al., 2006), but it also ensured that: 1) the selected voxels were defined objectively, 2) the ratio of voxels to trial repetitions in each ROI was reduced, decreasing the chance of classifier over-fitting (Pereira et al., 2009), 3) the voxels selected were statistically responsive to the task, reducing the chance of uninformative or noisy voxels being included in the ROI analysis, 4) ROI clusters were segregated from nearby activations (which is particularly important for parietal (i.e., SPOC, pIPS, midIPS, post. aIPS, aIPS) and lateral occipital areas (i.e., EBA and pMTG) which reside in close anatomical proximity and, at more liberal statistical thresholds, can have highly overlapping activations, 5) a similar number of voxels were included across frontoparietal cortex areas (from a minimum mean across subjects of 54 voxels in SPOC and PMv to a maximum mean across subjects of 89 voxels in motor cortex) and OTC areas (from a minimum mean across subjects of 23 voxels in pMTG to a maximum mean across subjects of 32 voxels in EBA), 6) the voxels included in each ROI were spatially contiguous, which may not be the case if voxel number was manipulated after cluster selection, and lastly, 7) a large enough number of voxels were included for spatial pattern classification.

We should also note that although the anatomical organization of frontoparietal cortex and lateral occipital cortex may be conducive to selecting and using *fewer* voxels from each region, it is not particularly conducive to including *more* voxels from each region. As well demonstrated by others, the lateral occipital cortex (where object-selective, face-selective, body-selective and tool-selective ROIs reside, the latter two areas localized and examined in the current study) contains several tightly spatially clustered category-selective ROIs, and their activations, at more liberal statistical thresholds, can be highly overlapping (e.g., Levy et al., 2001; Downing et al., 2007; Orlov et al., 2010). Thus, including additional voxels in each ROI (beyond what we have already selected, (15 mm)^3^) easily runs into the inherent problem that voxels being selected as belonging to the activity of one category-selective ROI, like EBA, become overlapping with the activity selected as part of a separate category-selective ROI, like pMTG (or some other ROI like the lateral occipital area, LO, or the occipital face area, OFA, both not localized in this study). Understandably, this would result in several interpretational issues for the data and this was a problem we wished to avoid in the current study by constraining the number of voxels included for analysis in each ROI to (15 mm)^3^ spatially contiguous clusters. Also, it is worth noting that when considering reducing the number of voxels in each ROI (to balance voxel number across ROIs), that the number of voxels presented in our analysis already represents, to a large extent, the lower end to the number that is typically reported in other studies.

Nevertheless, given that some other studies have shown that MVPA performance may be influenced by the number of voxels tested we have specifically addressed this issue by examining the impact of voxel number across our ROIs on the resulting decoding accuracies. In the figure below we plot decoding accuracy for each pair-wise comparison (Hand: G vs. R; Tool: G vs. R; Across-Effector: G vs. R) as a function of the number of voxels included in each ROI (13 ROIs total), where each data point in the plot denotes a single ROI (indicating the number of voxels for that ROI and its resulting pair-wise comparison decoding accuracy, as indicated by its colour and symbol). Lines of best fit, color-coded to correspond to a specific pair-wise comparison, are overlaid atop the data points, and demonstrate near zero correlations between both factors (voxel number and decoding accuracy). Importantly, none of the correlations for the three comparisons showed even a trend toward statistical significance. Thus, although other studies have shown that MVPA performance can be influenced by the number of voxels tested, that does not appear to be of concern for the range of cluster sizes included here. This finding validates our approach of selecting ROIs according to statistical criteria (t-test significance) and brain anatomy and, for many of the reasons already mentioned above, we believe it is problematic to directly balance the number of voxels (by arbitrarily selected fewer or more) across ROIs.Author response image 1Correlation between mean Voxel Number and resulting mean Decoding Accuracy (shown for each ROI (N=13) and each of the three pair-wise comparisons for the Plan-Epoch).Pearson correlation between Voxel Number and Hand Plan Epoch decoding accuracies (red diamond symbols): r^2^=0.108, p=0.272; Pearson Correlation between Voxel Number and Tool Plan Epoch decoding accuracies (blue square symbols): r^2^=0.025, p=0.602; Pearson Correlation between Voxel Number and Across-Effector Plan Epoch decoding accuracies (purple triangle symbols): r^2^=0.026, p=0.598.

*7) The authors averaged across both directions of the cross-effector test (hand→tool and tool→hand). It is possible however that there are asymmetries in action coding that would be revealed by disentangling these tests*.

This is an intriguing suggestion from the reviewers. We have directly addressed this comment by performing and plotting analyses that disentangle both directions of the cross-effector tests (Train set: Hand →Test set: Tool and Train set: Tool → Test set: Hand). These findings are now reported in Figure 4—figure supplement 1. As can be clearly seen in the figure, we found a largely symmetrical relationship between both directions of cross-effector classifier training and testing. We do realize that some researchers have previously reported asymmetrical differences in the direction of cross-decoding – and have made claims concerning such effects (e.g., Eger et al., 2009) – but we would be cautious to conclude that such effects are particularly meaningful and, moreover, whether they can actually be fully explained by underlying neural mechanisms. In binary pattern classification, all that is being learned by the classification algorithm is an optimal decision boundary that discriminates the spatial patterns of activity associated with two experimental conditions and it should be expected that if this decision boundary accurately generalizes to a completely different set of conditions (i.e., cross-decodes) and truly reflects a meaningful relationship in the data, that the particular direction of classifier training and testing should not reveal large differences in the resulting cross-classification accuracies.

Note that in the main-manuscript we have made reference to these supplemental analyses and we now write:

“[Note that separating these tests, Train set: Hand →Test set: Tool and Train set: Tool → Test set: Hand, revealed no major asymmetries in classification, see Figure 4—figure supplement 1].”

*8) While the authors find that the univariate response properties of regions generally correspond to decoding performance with MVPA, this could partly reflect the ROI approach that was used. A whole-brain MVPA mapping approach might well reveal other regions that code action preparation that are not activated (or not selectively activated) in the univariate sense*.

Although it is a fair point to note that our selection criteria of including frontoparietal ROIs involved in movement generation in our task (i.e., having higher activity for the Execute and Plan phases of the task than the Preview phase) perhaps makes it more likely that these areas will decode meaningful aspects of our motor task, we do find it worth noting that *the univariate response properties of these regions alone cannot account for the multitude of decoding profiles observed*. For instance, there is nothing in the signal amplitude time-course profiles of activity in SPOC or SMG that suggest that these areas will selectively code planned movements of the hand and tool, respectively (indeed, the same can be said for the OTC areas, EBA and pMTG). Likewise, there is nothing in the univariate response properties of areas aIPS and PMd to suggest that only the latter area will show accurate across-effector classification. These observations, of course, should not be particularly surprising given that our statistical selection criterion for voxel selection was fully orthogonal to the effects we report with MVPA. Indeed, as we wrote in the main manuscript: “… given the orthogonal contrast employed to select these 10 areas (i.e., Plan & Execute > Preview), their activity is not directionally biased to show any preview-, plan- or execute-related pattern differences between any of the experimental conditions.”

Indeed, it seems clear that if any strong interpretations are to be drawn from the voxel weight analyses performed in response to the reviewers’ suggestion #2, then it would be that successful decoding of planned movements cannot be explained by global differences in the response amplitudes across conditions but rather results from anisotropies in the distribution of voxel biases (see Figure 4—figure supplement 2 for confirmation of that fact).

Stated simply, the goal of the current study was to localize several key brain areas in frontoparietal and occipitotemporal cortex and determine what types of information they might represent related to the planning of hand- and tool-related actions. In the case of frontoparietal cortex, areas within this network have a long and well-documented history of being involved in the planning and control of hand actions and thus, from our perspective, it made perfect sense to localize and examine the planning-related activity in these areas in the context of our task. [Also, it should be noted that the frontoparietal areas included in the current manuscript are many of the exact same areas localized and analyzed in our previously published work (Gallivan et al., 2011a; Gallivan et al., 2011b; Gallivan et al., 2013) and it was of particular benefit for us as investigators to determine whether (and how) these same regions are involved in the planning of object-directed motor tasks involving an implement that is not actually part of the body, such as a tool]. In the case of OTC, several of these areas have a long and well-documented history of being involved in object-related perception and by independently localizing areas involved in body- and tool-related visual processing (i.e., EBA and pMTG, respectively) and extracting their activity from the motor-related task, we thought that it might be possible to reveal substantial insights into the roles that these areas play in generating hand- and tool-related actions (indeed, we believe that the matching visual-perceptual and motor-related functions of these areas observed in the current study does reveal such insight).

With regards to the reviewer comment concerning a whole-brain MVPA mapping approach, we are strongly in favour of providing readers with a well-reasoned hypothesis-driven ROI-based MVPA approach rather than providing largely post-hoc interpretations concerning the results of whole-brain MVPA maps (particularly in cases when strong predictions and hypotheses can be formed with regards to the involvement of particular areas in the task, as here). Moreover, we believe it is important to not gloss-over the fact that we already perform analyses and discuss, in quite significant detail, the activity of 13 different brain areas in our manuscript (and that number rises to 14, if the newly added auditory cortex findings are included in that tally). We are concerned that by extending our analysis (and related discussions) to areas not involved in the motor task (as indicated by their standard univariate responses for the contrast of Plan & Execute > Preview) nor body- or tool-selective visual processing (such as the independently localized areas, EBA and pMTG) that we will significantly weaken the focus and narrative of our paper. We should also note that some of the motivation for performing ROI-based MVPA here, as well as in our previous work in frontoparietal cortex (Gallivan et al., 2011a; Gallivan et al., 2011b; Gallivan et al., 2013), stems from the Whole-brain Searchlight MVPA approach requiring inter-subject anatomical averaging. In our experience, we have found that group stereotaxic averaging (e.g., Talairaching) performs quite poorly in many areas of parietal and frontal cortex. For example, group-activation in SPOC doesn’t survive intersubject averaging well despite there being highly robust effects at the single subject level; this is perhaps because SPOC is not only located far from the anatomical landmarks used for Talairach stereotaxic alignment but also due to the significant heterogeneity in the sulci of parietal cortex across participants (e.g., Quinlan and Culham, 2007).

In contrast, the ROI-based approach allows localization of brain activation with respect to anatomical landmarks in individuals, thus side-stepping these inherent problems of between-subject anatomical variability. Lastly, with respect to whole-brain MVPA, it should also be noted that performing decoding analysis for *each subject* (N=13) and for *each time-point* in the trial (i.e., 17 imaging volumes) across *each voxel of the brain* (∼50,000 separate searchlight clusters) presents inherent computational challenges that we are simply not equipped with the necessary computer hardware to deal with (at least within any reasonable time frame). In sum, we are happy to keep the paper more focused and manageable by presenting the findings from the 14 frontoparietal, auditory, and occipitotemporal cortex ROIs.

*9) The authors note the increased sensitivity of their method relative to a univariate approach, but MVPA (like any other method) carries assumptions and limitations that should shape how the data are interpreted. This could be discussed*.

The reviewers raise a very important point and indeed, we strongly believe that these assumptions and limitations should be adequately discussed in the context of interpreting our results. With this in mind, we have included such a discussion in the main manuscript.

As a final paragraph for our Results section, we now write:

“It is worth emphasizing that while accurate decoding in a region points to underlying differences in the neural representations associated with different experimental conditions (e.g., for reviews see Haynes and Rees, 2006; Norman et al., 2006; Kriegeskorte, 2011; Naselaris et al., 2011), a lack of decoding or ‘null effect’ (i.e., 50% chance classification) can either reflect that the region 1) is not recruited for the conditions being compared, 2) is similarly (but non-discriminately) engaged in those conditions, or 3) does in fact contain neural/pattern differences between the conditions but which cannot be discriminated by the pattern classification algorithm employed (i.e., a limit of methodology, see Pereira et al., 2009; Pereira and Botvinick, 2011). With respect to the first possibility, given that we selected frontoparietal cortex activity based on its involvement in the motor task at the single subject level (using the contrast of [Plan & Execute > Preview] across all conditions), it is reasonable to assume that all the localized areas are in some way engaged in movement generation. [Note that this general assumption is confirmed by the higher-than-baseline levels of activity observed in the signal amplitude responses during the Plan- and Execute-phases of the trial in areas of frontoparietal cortex (see Figures 2 and 5) and that this even appears to be the case in the independently localizer-defined lateral occipital cortex areas, EBA and pMTG (see Figure 6)]. Although it is understandably difficult to rule out the second possibility (i.e., that voxel pattern differences exist but are not detected with the SVM classifiers), it is worth noting that we do in fact observe null-effects with the classifiers in several regions where they are to be expected. For instance, SS-cortex is widely considered to be a lower-level sensory structure and anticipated to only show discrimination related to the motor task once the hand’s mechanoreceptors have been stimulated at object contact (either through the hand directly or through the tool, indirectly). Accordingly, here we find that SS-cortex activity only discriminates between grasp vs. reach movements following movement onset (i.e., during the Execute phase of the trial). Likewise, in motor cortex we show decoding for upcoming hand- and tool-related actions but, importantly, find no resulting across-effector classification. This latter result is highly consistent with the coding of differences in the hand kinematics required to operate the tool versus hand alone and accords with the presumed role of motor cortex in generating muscle-related activity (Kalaska, 2009; Churchland et al., 2012; Lillicrap and Scott, 2013). These findings in SS-cortex and motor cortex, when combined with the wide-range of decoding profiles found in other areas (i.e., from the hand-selective activity patterns in SPOC and EBA at one extreme, to the tool-selective activity patterns in SMG and pMTG at the other, for summary see Figure 7), suggest that the failure of some areas to decode information related to either hand- or tool-related trials (and not those of the other effector) is closely linked to invariance in the representations of those particular conditions. [To the extent that in cases where the activity of an area fails to discriminate between experimental conditions it can be said that the area is therefore not involved in coding (or invariant to) those particular conditions, we further expand upon interpretations related to these types of ‘null effects’ in the Discussion section.]”

*10) The authors report MVPA analyses at each time point, yet we did not find much in the discussion of the results about how these decoding effects develop over time*.

It is true that in the discussion of the results we do not examine in great detail how these decoding effects develop over time. In part, this was because we found very high correspondence between the Plan-Epoch decoding results and those of the time-resolved decoding analysis (i.e., single time-point results). Nevertheless, we do feel it important to note that these “time-point” decoding results provide three rather general but significant observations. We have now noted these important observations in the main manuscript and we write:

“Three general observations can be made based on the results of these decoding analyses. First, predictive movement information, if it is to emerge, generally arises in the two time points prior to initiation of the movement (although note that in a few brain areas, such as L-pIPS and L-PMd, this information is also available prior to these two time points). Second, in support of the notion that this predictive motor information is directly related to the *intention* to make a movement, accurate classification never arises prior to the subject being aware of which action to execute (i.e., prior to the auditory instruction delivered at the initiation of the Plan phase). Finally, decoding related to the planning of a movement can be fully disentangled from decoding related to movement execution, which generally arises several imaging volumes later.”

As a brief aside, we find it important to note that these time-resolved decoding results also provide an validation of our past published work (Gallivan et al., 2011a; Gallivan et al., 2011b; Gallivan et al., 2013), which divided the trial into separate epochs for decoding analysis (i.e., Preview, Plan and Execute).

**References**

Bracci S, Cavina-Pratesi C, Ietswaart M, Caramazza A, Peelen MV (2012) Closely overlapping responses to tools and hands in left lateral occipitotemporal cortex. Journal of neurophysiology 107:1443-1456.

Churchland MM, Cunningham JP, Kaufman MT, Foster JD, Nuyujukian P, Ryu SI, Shenoy KV (2012) Neural population dynamics during reaching. Nature 487:51-56.

Cisek P (2007) Cortical mechanisms of action selection: the affordance competition hypothesis. Phil Transact R Soc Lond B 362:1585-1599.

Cisek P, Kalaska JF (2010) Neural Mechanisms for Interacting with a World Full of Action Choices. Annu Rev Neurosci.

Downing PE, Wiggett AJ, Peelen MV (2007) Functional magnetic resonance imaging investigation of overlapping lateral occipitotemporal activations using multi-voxel pattern analysis. The Journal of neuroscience : the official journal of the Society for Neuroscience 27:226-233.

Downing PE, Chan AW, Peelen MV, Dodds CM, Kanwisher N (2006) Domain specificity in visual cortex. Cerebral cortex 16:1453-1461.

Eger E, Michel V, Thirion B, Amadon A, Dehaene S, Kleinschmidt A (2009) Deciphering cortical number coding from human brain activity patterns. Current biology: CB 19:1608-1615.

Gallivan JP, McLean DA, Smith FW, Culham JC (2011a) Decoding effector-dependent and effector-independent movement intentions from human parieto-frontal brain activity. Journal of Neuroscience 31:17149-17168.

Gallivan JP, McLean DA, Flanagan JR, Culham JC (2013) Where one hand meets the other: limb-specific and action-dependent movement plans decoded from preparatory signals in single human frontoparietal brain areas. The Journal of neuroscience : the official journal of the Society for Neuroscience 33:1991-2008.

Gallivan JP, McLean DA, Valyear KF, Pettypiece CE, Culham JC (2011b) Decoding action intentions from preparatory brain activity in human parieto-frontal networks. Journal of Neuroscience 31:9599-9610.

Harrison SA, Tong F (2009) Decoding reveals the contents of visual working memory in early visual areas. Nature 458:632-635.

Haxby JV, Gobbini MI, Furey ML, Ishai A, Schouten JL, Pietrini P (2001) Distributed and overlapping representations of faces and objects in ventral temporal cortex. Science 293:2425-2430.

Haynes JD, Rees G (2006) Decoding mental states from brain activity in humans. Nature reviews Neuroscience 7:523-534.

Jeannerod M, Arbib MA, Rizzolatti G, Sakata H (1995) Grasping objects: the cortical mechanisms of visuomotor transformation. Trends in Neurosciences 18:314-320.

Kalaska JF (2009) From intention to action: motor cortex and the control of reaching movements. Adv Exp Med Biol 629:139-178.

Kamitani Y, Tong F (2005) Decoding the visual and subjective contents of the human brain. Nature neuroscience 8:679-685.

Kanwisher N (2010) Functional specificity in the human brain: a window into the functional architecture of the mind. Proceedings of the National Academy of Sciences of the United States of America 107:11163-11170.

Kriegeskorte N (2011) Pattern-information analysis: From stimulus decoding to computational-model testing. NeuroImage 56:411-421.

Levy I, Hasson U, Avidan G, Hendler T, Malach R (2001) Center-periphery organization of human object areas. Nature neuroscience 4:533-539.

Lillicrap TP, Scott SH (2013) Preference distributions of primary motor cortex neurons reflect control solutions optimized for limb biomechanics. Neuron 77:168-179.

Mahon BZ, Caramazza A (2009) Concepts and categories: a cognitive neuropsychological perspective. Annu Rev Psychol 60:27-51.

Mahon BZ, Caramazza A (2011) What drives the organization of object knowledge in the brain? Trends in cognitive sciences 15:97-103.

Mahon BZ, Milleville SC, Negri GA, Rumiati RI, Caramazza A, Martin A (2007) Action-related properties shape object representations in the ventral stream. Neuron 55:507-520.

Murata A, Gallese V, Luppino G, Kaseda M, Sakata H (2000) Selectivity for the shape, size, and orientation of objects for grasping in neurons of monkey parietal area AIP. Journal of neurophysiology 83:2580-2601.

Naselaris T, Kay KN, Nishimoto S, Gallant JL (2011) Encoding and decoding in fMRI. NeuroImage 56:400-410.

Norman KA, Polyn SM, Detre GJ, Haxby JV (2006) Beyond mind-reading: multi-voxel pattern analysis of fMRI data. Trends in cognitive sciences 10:424-430.

Orlov T, Makin TR, Zohary E (2010) Topographic representation of the human body in the occipitotemporal cortex. Neuron 68:586-600.

Pereira F, Botvinick M (2011) Information mapping with pattern classifiers: A comparative study. NeuroImage.

Pereira F, Mitchell T, Botvinick M (2009) Machine learning classifiers and fMRI: a tutorial overview. NeuroImage 45:S199-209.

Quinlan DJ, Culham JC (2007) fMRI reveals a preference for near viewing in the human parieto-occipital cortex. NeuroImage 36:167-187.

Rizzolatti G, Luppino G (2001) The cortical motor system. Neuron 31:889-901.